# Provable Subspace Identification Under Post-Nonlinear Mixtures

**Qi Lyu**
School of EECS
Oregon State University
Corvallis, OR 97331
`lyuqi@oregonstate.edu`

**Xiao Fu**
School of EECS
Oregon State University
Corvallis, OR 97331
`xiao.fu@oregonstate.edu`

## Abstract

Unsupervised mixture learning (UML) aims at identifying linearly or nonlinearly mixed latent components in a blind manner. UML is known to be challenging: Even learning linear mixtures requires highly nontrivial analytical tools, e.g., independent component analysis or nonnegative matrix factorization. In this work, the post-nonlinear (PNL) mixture model—where *unknown* element-wise nonlinear functions are imposed onto a linear mixture—is revisited. The PNL model is widely employed in different fields ranging from brain signal classification, speech separation, remote sensing, to causal discovery. To identify and remove the unknown nonlinear functions, existing works often assume different properties on the latent components (e.g., statistical independence or probability-simplex structures). This work shows that under a carefully designed UML criterion, the existence of a nontrivial *null space* associated with the underlying mixing system suffices to guarantee identification/removal of the unknown nonlinearity. Compared to prior works, our finding largely relaxes the conditions of attaining PNL identifiability, and thus may benefit applications where no strong structural information on the latent components is known. A finite-sample analysis is offered to characterize the performance of the proposed approach under realistic settings. To implement the proposed learning criterion, a block coordinate descent algorithm is proposed. A series of numerical experiments corroborate our theoretical claims.

## 1 Introduction

Unsupervised mixture learning (UML) has been broadly used in machine learning tasks, e.g., for recovering mixed latent components or extracting informative data representations. Central to understanding UML is the concept of *identifiability*. The identifiability of UML is concerned with whether the latent components can be identified (up to certain to inconsequential ambiguities) with guarantees—without using any label or training data. This is a challenging problem due to the unsupervised nature. To address the identifiability challenge, linear mixture learning problems were tackled via exploiting properties of the latent components, e.g., statistical independence [1], nonnegativity [2–4], quasi-stationarity [5,6], and boundedness [7]. This leads to a cohort of *blind source separation* (BSS) methods.

However, linear mixture models (LMMs) oftentimes fail to perform well on real-world data, due to the unknown nonlinear distortions arising in data generation and acquisition. In the past two decades, nonlinear mixture models (NMMs) have attracted unprecedentedly more attention [8–14]. Nonetheless, the identifiability of the latent components is much harder to establish under NMMs. It was shown in [15] that the NMMs are in general not identifiable, even with strong assumptions, e.g., statistical independence among the latent components. To establish NMM identifiability, a line of work exploited observable auxiliary variables or specific priors (i.e., extra information about the data

or the latent components on top of statistical independence); see, e.g., [13,14,16,17]. Another line of methods took an alternative route via utilizing the structures of the nonlinear distortions. In particular, the *post-nonlinear mixture* (PNL) model [8,9,18–20] is a reasonable extension of the classical LMM by assuming *unknown* nonlinear transformations *individually* imposed onto the outputs of a underlying linear mixture. This kind of "structured NMM" is used in a wide range of applications, e.g. hyperspectral imaging [19,21], brain data classification [20,22–24], and causal discovery [12,25,26]. In order to identify the generative model/latent components under the PNL model, a key step is to remove the nonlinear distortions in an unsupervised manner. To this end, the existing methods often explicitly rely on some properties of the latent components, e.g., statistical independence [8,9,18], availability of multiview data [20] and the sum-to-one and nonnegativity (or probability-simplex) structure of the latent components [19,27]. These structures stem from specific applications, and thus are meaningful—yet not always available. A natural question is whether the nonlinear distortions are still removable if the widely used structural assumptions (e.g., those in [8,9,18–20,27]) do not hold?

In this work, we offer a positive answer to the above inquiry—and build a new learning algorithm based on our analyses. Our detailed contributions are as follows.

**Subspace-Based PNL Model Identification.** We propose a new nonlinearity identification and removal criterion under the PNL model. We show that, as long as the underlying *unknown* linear mixing system of the PNL model admits a nontrivial orthogonal complement (or, its transpose has a nontrivial null space), the unknown nonlinear transformations are identifiable and removable, under mild conditions. Notably, our approach does not use any specific structural assumptions of the latent components. Our finding presents a fairly different result compared to existing nonlinear mixture learning techniques. The latter often heavily rely on the properties of the latent components. Hence, our result may offer new insights and a useful alternative to this long-standing challenge. An immediate consequence is that our method can work with various types of latent components, including those not covered by existing PNL model identification approaches in [8,9,18–20,27].

**Performance Analysis Under Realistic Settings.** The model identification analysis is conducted under the population case where unlimited data is available, with the assumption that a universal function learner can be used to cancel any nonlinear distortion. To advance the understanding under more realistic settings, we also consider the realistic scenario where one only has access to finite samples and non-universal function learners. The analysis is a combination or statistical generalization theory and numerical differentiation, which is reminiscent of the recently developed framework in [20,28,29]—with nontrivial additional efforts to accommodate our learning criterion.

**Optimization Algorithm Design.** Based on the theoretical analysis, we formulate the identification problem as a constrained optimization criterion. The optimization problem is challenging, as it involves learning unknown nonlinear functions [represented by neural networks (NNs)] under nonconvex constraints. We propose a block coordinate descent (BCD) algorithm that alternates between two subproblems, namely, null space basis learning and nonlinear function learning. This way, the latter is disentangled with constraints, and thus off-the-shelf NN training techniques (e.g., Adam [30]) can be readily employed to handle the subproblem. A series of experiments are conducted for validating our theoretical claims.

**Notations.** $x, \boldsymbol{x}$, and $\boldsymbol{X}$ represent a scalar, a vector, and a matrix, respectively; $\|\boldsymbol{x}\|_0$, $\|\boldsymbol{x}\|_1$, and $\|\boldsymbol{x}\|_2$ denote the $\ell_0$ function, the $\ell_1$ norm, and the Euclidean norm, respectively; $\odot$ and $\circledast$ denote the Khatri-Rao product and the Hadamard product, respectively; $f$ is used to denote a function $f(\cdot) : \mathbb{R} \to \mathbb{R}$; $f'$, $f''$ and $f^{(n)}$ denote the first-order, the second-order, and the $n$th order derivatives of the function $f$, respectively; $f \circ g$ denotes the function composition; $\mathcal{R}(\boldsymbol{X})$ and $\mathcal{N}(\boldsymbol{X})$ mean the range space and the null space of $\boldsymbol{X}$, respectively.

## 2 Background

### 2.1 UML and Model Identification

In UML, the classical LMM is defined as follows [31]:

$$\boldsymbol{x}_\ell = \boldsymbol{A}\boldsymbol{s}_\ell, \ \ell = 1, 2, \cdots, N, \tag{1}$$

where $\boldsymbol{x}_\ell \in \mathbb{R}^M$ is the $\ell$th observation sample, $\boldsymbol{s}_\ell \in \mathbb{R}^K$ contains the $K$ latent components, and $\boldsymbol{A} \in \mathbb{R}^{M \times K}$ is the mixing system. The latent components in (1) are generally not identifiable,

if $\boldsymbol{A}$ is not known [31]. As a remedy, information about $\boldsymbol{A}$ and/or $\boldsymbol{s}_\ell$ is often used to establish identifiability. For instance, statistical independence among $[\boldsymbol{s}_\ell]_i$'s is used in independent component analysis (ICA) [1], and the nonnegativity of $\boldsymbol{A} \in \mathbb{R}_+^{M \times K}$ and $\boldsymbol{S} = [\boldsymbol{s}_1, \cdots, \boldsymbol{s}_N] \in \mathbb{R}_+^{K \times N}$ are used in nonnegative matrix factorization (NMF) [3,4]; also see [5–7,32–34] for more conditions that can assist LMM identification.

The LMM in (1) is often over-simplified for modeling the data generation/acquisition processes in real-world applications—where nonlinear effects are widely observed. In many cases, the NMMs are of more interest. A typical NMM is as follows [35]:

$$\boldsymbol{x}_\ell = \boldsymbol{g}(\boldsymbol{s}_\ell),\ \ell = 1, 2, \cdots, N, \tag{2}$$

where $\boldsymbol{g} : \mathbb{R}^K \to \mathbb{R}^M$ is a nonlinear "mixing system". Compared to the LMM, the NMM in (2) can represent much more complex data generation/acquisition processes. However, it was shown in [35] that the model in (2) is not identifiable, even if the latent components satisfy some fairly strong assumptions, e.g., statistical independence. A workaround is to use more information of or additional conditions on the latent components, e.g., the availability of auxiliary variables [13,14,16]. Another approach is by exploiting structural assumptions of the nonlinear mixing process. This leads to the suite of PNL model learning works (see, e.g., [8,9,18–20,27,36]), which will be our focus.

## 2.2 Post-Nonlinear Mixture Model

The PNL model is a natural extension to the LMM, which is expressed as follows [8,9,18,20,27,36]:

$$\boldsymbol{x}_\ell = \boldsymbol{g}(\boldsymbol{A}\boldsymbol{s}_\ell),\ \ell = 1, 2, \cdots, N, \tag{3}$$

where $\boldsymbol{g} = [g_1(\cdot), \cdots, g_M(\cdot)]^\top$ with each $g_m(\cdot) : \mathbb{R} \to \mathbb{R}$ being an invertible nonlinear function. We assume $\boldsymbol{s}_\ell \sim p(\boldsymbol{s})$ and the support of $p(\boldsymbol{s})$ is a continuous and open domain $\mathcal{S}$.

The PNL model has a myriad of of applications across different fields, e.g., causal discovery [12,25,26], hyperspectral imaging [21] and brain data classification [22–24]. It is considered as an appropriate model when the unknown nonlinear distortions happen at the sensor end in data acquisition. It also offers a valuable generalization to the linear mixture generative models in data analytics, e.g., those used in NMF and data clustering [37,38].

An important remark is that *the challenge of UML under the PNL model boils down to identifying and removing $\boldsymbol{g}(\cdot)$*, as this will lead to a well-understood LMM as in (1). Then, any LMM identification approach, e.g., those in [3,6,7,31,32,34,39], can be applied to identify the latent components $\boldsymbol{s}_\ell$ for $\ell = 1, \ldots, N$.

Existing works tackle the $\boldsymbol{g}$-identification and removal problem from various angles. In [8,18], the statistical independent among the latent components (i.e., $[\boldsymbol{s}_\ell]_k$ for $k = 1, \ldots, K$) is used. Recently, [19,27] proposed to identify the PNL model with latent components satisfying the probability simplex constraint (i.e., $\mathbf{1}^\top \boldsymbol{s}_\ell = 1$, $\boldsymbol{s}_\ell \geq \mathbf{0}$) that often arises in soft clustering problems. The work in [20] shows that $\boldsymbol{g}(\cdot)$ can be provably removed if multiple views of data exist. These methods offered insightful and useful approaches to remove $\boldsymbol{g}(\cdot)$, but the use of specific structural assumptions on the latent components make their applicability relatively narrow.

We propose to identify and remove the nonlinear distortions via exploiting an underlying subspace that often naturally exists under PNL models, instead of resorting to structural assumptions of $\boldsymbol{s}_\ell$ as in prior works. Our method starts with the following observation: If the mixing system $\boldsymbol{A}$ is a tall matrix, $\mathcal{N}(\boldsymbol{A}^\top)$ is a nontrivial subspace. This implies that if only the linear mixture part is considered (i.e., if $\boldsymbol{g}(\cdot)$ can be removed completely), there exist vectors $\boldsymbol{v}$ such that

$$\boldsymbol{v}^\top \boldsymbol{A}\boldsymbol{s}_\ell = 0,\ \forall \ell, \tag{4}$$

where the nonzero $\boldsymbol{v}$ is any vector from $\mathcal{N}(\boldsymbol{A}^\top)$. In the next section, we will show how this simple equation in (4) allows for constructing a provable $\boldsymbol{g}$-identification and removal criterion.

## 3 Proposed Approach

In this section, we propose a learning criterion to identify/remove the nonlinear transformations $\boldsymbol{g}(\cdot)$. We first consider the population case where infinite data points are available; that is, all $\boldsymbol{x}_\ell \sim p(\boldsymbol{x})$ over the continuous support $\mathcal{X}$ are available. Then, we will move forward to consider the finite-sample case in Sec. 3.2. Under the infinite-data assumption, our $\boldsymbol{g}$-identification formulation is as follows:

$$\text{find } \boldsymbol{Q}, \ \boldsymbol{f}(\cdot) = [f_1(\cdot), \cdots, f_M(\cdot)]^\top \tag{5a}$$

$$\text{s.t. } \boldsymbol{Q}^\top \boldsymbol{f}(\boldsymbol{x}_\ell) = \boldsymbol{0}, \ \forall \boldsymbol{x}_\ell \in \mathcal{X} \tag{5b}$$

$$f_m(\cdot) : \mathbb{R} \to \mathbb{R} \text{ is invertible } \forall m, \tag{5c}$$

$$\|\boldsymbol{Q}\|_0 = MD, \ \boldsymbol{Q}^\top \boldsymbol{Q} = \boldsymbol{I}, \tag{5d}$$

where $\boldsymbol{Q} \in \mathbb{R}^{M \times D}$, $D$ is the dimension of the null space (i.e., $\mathcal{N}(\boldsymbol{A}^\top)$), in which $D \geq 1$ and $D < M$. The $\ell_0$ function constraint in (5d) means that $\boldsymbol{Q}$ is constrained to be a dense matrix, which will prove important for establishing identifiability; this will become clearer in Sec. 3.1. In the formulation, $f_m(\cdot)$ is a nonlinear invertible function that we look for. Ideally, we hope to learn $f_m(\cdot) = g_m^{-1}(\cdot)$ which cancels the nonlinear transformation. Then, $\boldsymbol{Q}$ forms a basis of $\mathcal{N}(\boldsymbol{A}^\top)$. Note that under the orthogonality constraint in (5d), the trivial solution, i.e., $\boldsymbol{Q} = \boldsymbol{0}$, is avoided. Besides, the invertibility constraint in 5c is used to prevent other degenerate solutions, e.g., $\boldsymbol{f}(\boldsymbol{x}_\ell) = \boldsymbol{c}$ for all $\ell$ with a constant $\boldsymbol{c}$, from happening.

### 3.1 Provable Nonlinearity Removal

We will show that solving (5) guarantees that the nonlinear transformation $\boldsymbol{g}(\cdot)$ is removed from the PNL model. To this end, we first introduce the following condition:

**Definition 1 (Locally Free Components)** *Consider real-valued random vector $\boldsymbol{v} = [v_1, \ldots, v_d]^\top \in \mathbb{R}^d$, where $v_i$ resides in a continuous and open set $\mathcal{V}_i \subseteq \mathbb{R}$. Assume that the following holds for all $j$*

$$p(v_j | \boldsymbol{v}_{-j}) > 0, \ \text{with continuous support } \mathcal{V}_{j|\boldsymbol{v}_{-j}} \text{ given any } \boldsymbol{v}_{-j},$$

*where $\boldsymbol{v}_{-j}$ denotes the vector with all entries but $v_j$. Then, the components in $\boldsymbol{v}$ are all locally free components.*

Note that the definition above means that each $v_j$ has the "freedom" to change (at least locally) no matter what value the $\boldsymbol{v}_{-j}$ part takes. As a special case, statistically independent random variables clearly satisfy the definition. However, statistical independence is a much stronger condition. Dependent variables could also be locally free components. For instance, vectors from the probability simplex, i.e.,

$$\boldsymbol{v} \in \boldsymbol{\Delta}_K, \ \boldsymbol{\Delta}_K = \{\boldsymbol{v} \in \mathbb{R}^K | \boldsymbol{1}^\top \boldsymbol{v} = 1, \boldsymbol{v} > \boldsymbol{0}\}, \tag{6}$$

also satisfy Definition 1—that is, any $K - 1$ of the $K$ components in $\boldsymbol{v}$ are locally free components. Generally, if $v_1, \ldots, v_d$ are not completely dependent, the locally free condition is not hard to meet.

Simply speaking, with $v_i$ and $v_j$ being free variables, we could observe any possible value of $v_j$ given a fixed $\bar{v}_i$. Using this notion, consider the functional equation $\boldsymbol{q}^\top \boldsymbol{h}(\boldsymbol{A}\boldsymbol{v}) = 0$ that holds for all $\boldsymbol{v}$. Then, we can always observe

$$\boldsymbol{q}^\top \boldsymbol{h} \left( \boldsymbol{A}[\bar{v}_1, \cdots, v_j, \cdots, \bar{v}_d]^\top \right) = 0, \tag{7}$$

with any fixed $\bar{\boldsymbol{v}}_{-j}$, where $v_j$ is from the continuous support of $p(v_j | \bar{\boldsymbol{v}}_{-j}) > 0$. Taking derivative of (7) w.r.t. $v_j$ leads to the following:

$$(\boldsymbol{q}^\top \odot \boldsymbol{a}_j^\top) \boldsymbol{h}' \left( \boldsymbol{A}[\bar{v}_1, \cdots, v_j, \cdots, \bar{v}_d]^\top \right) = 0, \tag{8}$$

as all the terms $\partial \bar{v}_i / \partial v_j = 0$ for any $i \neq j$. This property will help us show the following main result:

**Theorem 1 (Nonlinearity Identification and Removal)** *Under the model in (3), assume that the criterion (5) is solved. In addition, assume that $\boldsymbol{A} \in \mathbb{R}^{M \times K}$ is drawn from any joint absolutely continuous distribution, that $\widetilde{K}$ of the $K$ components of $\boldsymbol{s}$ are locally free components (cf. Def. 1), and that the learned $\widehat{h}_m = \widehat{f}_m \circ g_m$ is twice differentiable for all $m \in [M]$. Suppose that*

$$\frac{\widetilde{K}(\widetilde{K} + 1)}{2} \geq M. \tag{9}$$

*Then, almost surely, any $\widehat{\boldsymbol{f}}(\cdot)$ that is a solution of (5) satisfies $\widehat{h}_m(x) = \widehat{f}_m \circ g_m(x) = c_m x + d_m$, $\forall m \in [M]$, where $c_m \neq 0$ and $d_m$ is a constant, at the limit of infinite data.*

Theorem 1 indicates that if any solution $\widehat{\boldsymbol{f}}(\cdot)$ of (5) is found, then the composition $\widehat{h}_m = \widehat{f}_m \circ g_m$ is an affine function for all $m$. Notably, the theorem does not require all the $K$ latent components to be locally free components, as long as the condition $\widetilde{K}(\widetilde{K}+1)/2 \geq M$ is satisfied—which means that some components are even allowed to be completely dependent. Intuitively, if $\widetilde{K}$ is too small then it means that the remaining $K - \widetilde{K}$ components do not provide useful information, while a larger $\widetilde{K}$ implies that there is more diversity among the latent components to assist nonlinearity removal. Note that the premise here is that the null space $\mathcal{N}(\boldsymbol{A}^\top)$ must exist, i.e., $M > K$ should hold as $\text{rank}(\boldsymbol{A}) = \min\{M, K\}$ with probability one for any $\boldsymbol{A}$ that is drawn from a joint continuous distribution. Otherwise, one may fail to learn a nonzero $\boldsymbol{Q}$. On the other hand, if $M$ is overly large, i.e., $M > \widetilde{K}(\widetilde{K}+1)/2$, the model is not identifiable as there is a lack of information (i.e., the number of locally free components is not large enough) to pin down all the $g_i(\cdot)$'s with $i = 1, \cdots, M$. We show the proof as follows:

**Proof**: By solving (5), we have the following equation for all $\boldsymbol{s}_\ell \in \mathcal{S}$

$$\boldsymbol{Q}^\top \boldsymbol{h}(\boldsymbol{A}\boldsymbol{s}_\ell) = \boldsymbol{0}, \tag{10}$$

where $\boldsymbol{h} = \boldsymbol{f} \circ \boldsymbol{g}$ is the function composition. For simplicity, we drop the subscript "$\ell$" as the equation holds for any $\boldsymbol{s} \in \mathcal{S}$.

First, we show the existence of solution to (5). It is obvious that one can let $\boldsymbol{f} = \boldsymbol{g}^{-1}$ (i.e., $f_m = g_m^{-1}$ for all $m$) and make the columns of $\boldsymbol{Q}$ to be an orthogonal basis of $\mathcal{N}(\boldsymbol{A}^\top)$. In addition, such a $\boldsymbol{Q}$ can always be made dense. This is because $\boldsymbol{A}$ follows a joint continuous distribution, which indicates that the probability of $\mathcal{N}(\boldsymbol{A}^\top)$ being orthogonal to any axis is zero.

Next, we show that such a solution is unique up to only affine ambiguities. By the assumptions, there are $\widetilde{K}$ out of the $K$ components of $\boldsymbol{s}$ that are locally free. Thus, by taking the second-order (cross) derivatives of Eq. 10 w.r.t. each of the $\widetilde{K}$ variables, as shown in (8), we have the following:

$$\boldsymbol{Q}^\top \odot \underbrace{\begin{bmatrix} (\boldsymbol{a}_1 \circledast \boldsymbol{a}_1)^\top \\ \vdots \\ (\boldsymbol{a}_{\widetilde{K}} \circledast \boldsymbol{a}_{\widetilde{K}})^\top \\ (\boldsymbol{a}_1 \circledast \boldsymbol{a}_2)^\top \\ \vdots \\ \left(\boldsymbol{a}_{\widetilde{K}-1} \circledast \boldsymbol{a}_{\widetilde{K}}\right)^\top \end{bmatrix}}_{\boldsymbol{B}} \begin{bmatrix} \widehat{h}_1''([\boldsymbol{A}\boldsymbol{s}]_1) \\ \vdots \\ \widehat{h}_M''([\boldsymbol{A}\boldsymbol{s}]_M) \end{bmatrix} = \boldsymbol{0}, \ \forall \boldsymbol{s} \in \mathcal{S}, \tag{11}$$

where $\boldsymbol{a}_k$ is the $k$th column of $\boldsymbol{A}$. Without loss of generality, we have assumed that the first $\widetilde{K}$ variables of $\boldsymbol{s}$ are locally free components.

Note that if one can show $\widehat{h}_m''([\boldsymbol{A}\boldsymbol{s}]_m) = 0$ for any $m$ over any $\boldsymbol{s}$, then it indicates that $\widehat{h}_m(\cdot)$ is an affine function for all $m$ over $\mathcal{S}$. To this end, we show that $\boldsymbol{Q}^\top \odot \boldsymbol{B}$ has full column rank, where the sizes of the matrices involved are $\boldsymbol{Q} \in \mathbb{R}^{M \times D}$ and $\boldsymbol{B} \in \mathbb{R}^{\frac{\widetilde{K}(\widetilde{K}+1)}{2} \times M}$.

By Lemma 1 of [40], for matrices $\boldsymbol{U} \in \mathbb{R}^{a \times m}$ and $\boldsymbol{V} \in \mathbb{R}^{b \times m}$, $\boldsymbol{U} \odot \boldsymbol{V}$ has full column rank if $\text{krank}(\boldsymbol{U}) + \text{krank}(\boldsymbol{V}) \geq m + 1$ with $\text{krank}(\boldsymbol{U})$ and $\text{krank}(\boldsymbol{V})$ being the Kruskal rank of $\boldsymbol{U}$ and $\boldsymbol{V}$, respectively. The Kruskal rank of matrix $\boldsymbol{U}$ is defined as the largest number $\tau$ such that every set of $\tau$ columns of $\boldsymbol{U}$ is linearly independent. For $\boldsymbol{Q}^\top \odot \boldsymbol{B}$, we do not have control over $\boldsymbol{Q}$ since it is an optimization variable. But the Kruskal rank $k_{\boldsymbol{Q}^\top}$ is at least 1 since we have the constraint $\|\boldsymbol{Q}\|_0 = MD$. In terms of $\boldsymbol{B}$, we can show that the matrix $\boldsymbol{B}$ has full Kruskal rank $\min\left(\widetilde{K}(\widetilde{K}+1)/2, M\right)$, almost surely. The detailed explanation can be found in Appendix B.

To derive the conditions required for model identification (i.e., $\text{rank}(\boldsymbol{Q}^\top \odot \boldsymbol{B}) = M$), we consider the following two cases:

a) For the case of $\text{krank}(\boldsymbol{B}) = \frac{\widetilde{K}(\widetilde{K}+1)}{2}$, $\text{rank}(\boldsymbol{Q}^\top \odot \boldsymbol{B}) = M$ with probability one when the following conditions are met:

$$M \geq \frac{\widetilde{K}(\widetilde{K}+1)}{2} \quad \text{and} \quad 1 + \frac{\widetilde{K}(\widetilde{K}+1)}{2} \geq M + 1, \text{ which holds if } M = \frac{\widetilde{K}(\widetilde{K}+1)}{2}.$$

b) For the case of $\mathrm{krank}(\boldsymbol{B}) = M$, $\mathrm{rank}(\boldsymbol{Q}^\top \odot \boldsymbol{B}) = M$ with probability one when the following conditions are met:

$$M \leq \frac{\widetilde{K}(\widetilde{K}+1)}{2} \quad \text{and} \quad 1 + M \geq M + 1, \text{ which holds if } \frac{\widetilde{K}(\widetilde{K}+1)}{2} \geq M.$$

The two cases imply that if $\frac{\widetilde{K}(\widetilde{K}+1)}{2} \geq M$ is satisfied, then the nonlinear functions $g_m(\cdot)$'s can be removed almost surely. ∎

## 3.2 Finite-Sample Identifiability Analysis

The identifiability analysis in Sec. 3.1 is based on the assumption that (uncountably) infinite samples are available. However, one always has to work with finite samples in practice. In addition, the proof of Theorem 1 assumed that the $f_m(\cdot)$'s are universal function approximators. Nonetheless, in practice, the function learners, e.g., neural networks, may have nontrivial approximation errors for modeling nonlinear functions. In this section, we provide analysis under more realistic conditions.

To begin with, note that when only $N$ i.i.d. samples are available, one could rewrite the finite-sample version of (5) as follows

$$\min_{\boldsymbol{Q},\boldsymbol{f}} \frac{1}{N} \sum_{\ell=1}^{N} \left\| \boldsymbol{Q}^\top \boldsymbol{f}(\boldsymbol{x}_\ell) \right\|_2^2 \tag{12a}$$

$$\text{s.t. } \|\boldsymbol{Q}\|_0 = MD, \ \boldsymbol{Q}^\top \boldsymbol{Q} = \boldsymbol{I}, \ f_m(\cdot) \in \mathcal{F}, \tag{12b}$$

where $\mathcal{F} : \mathbb{R} \to \mathbb{R}$ is an invertible function class used for modeling the inverse of function $g_m(\cdot)$. In the population case (i.e., $N = \infty$), (12) and (5) have the same solutions. To characterize the finite sample performance of (12) under $\boldsymbol{f}(\cdot)$ that has limited expressive power, we will use the following definition and assumptions:

**Definition 2 (Function Class and Inverse)** *The function $g \in \mathcal{G}$ is an invertible continuous nonlinear function. We define its inverse class $\mathcal{G}^{-1}$ as a function class that contains all the $u$'s satisfying $u \circ g(y) = \alpha y + \beta$ with $\alpha \neq 0$, $\forall g \in \mathcal{G}$.*

**Assumption 1 (Realization Gap)** *For any $u \in \mathcal{G}^{-1}$, there exists $f \in \mathcal{F}$ such that $\sup_{\boldsymbol{x} \in \mathcal{X}} |f(x_m) - u(x_m)| \leq \nu$, $\forall m \in [M]$.*

**Assumption 2 (Boundedness)** *The 4th-order derivatives of $f_m \in \mathcal{F}$ and $g_m \in \mathcal{G}$ exist. In addition, $|f_m^{(n)}(\cdot)|$ and $|g_m^{(n)}(\cdot)|$ are bounded for all $n \in [4]$ and $m \in [M]$. Any 4th-order derivative of $\boldsymbol{q}_k^\top \boldsymbol{h}(\boldsymbol{A}\boldsymbol{s}_\ell)$ with $k = [D]$ is bounded by $C_d$. In addition, $\boldsymbol{A}$ has bounded elements.*

**Assumption 3 (Neural Network Structure)** *The function $f_m(\cdot)$ is parameterized by a one-hidden-layer neural network with $R$ neurons and 1-Lipschitz nonlinear activation function $\zeta(\cdot) : \mathbb{R} \to \mathbb{R}$ with $\zeta(0) = 0$[1].*

$$\mathcal{F} = \{f_m | f_m(x) = \boldsymbol{w}_2^\top \boldsymbol{\zeta}(\boldsymbol{w}_1 x), \|\boldsymbol{w}_i\|_2 \leq B, \ i = 1, 2\}, \tag{13}$$

*where $\boldsymbol{\zeta}(\boldsymbol{y}) = [\zeta(y_1), \ldots, \zeta(y_R)]^\top$, $\boldsymbol{w}_i \in \mathbb{R}^R$ for $i = 1, 2$.*

Next, we show the finite-sample identifiability result for the case where the above neural network class is used. Our analysis can be readily generalized to cover more complex neural networks such as deep convolutional neural network (CNN) and ResNet. However, we use this simple neural network as a showcase, since our goal is to reveal insights other than covering complex neural structures.

Using the neural networks from Assumption 3, we have the following lemma:

---

[1]Note that commonly used activation functions, e.g., , tanh, rectified linear unit (ReLU) and exponential linear unit (ELU), all satisfy this condition; see [41].

**Lemma 1** *Assume that $\mathcal{F}$ is the function class defined in Assumption 3 and the input $\boldsymbol{x}$ is bounded by $|x_m| \leq C_x$ for all $m$. Then, the loss function $\left\|\boldsymbol{Q}^\top \boldsymbol{f}(\boldsymbol{x}_\ell)\right\|_2^2$ has the following Rademacher complexity*

$$\Re \leq 2DMB^4 C_x^2 \sqrt{\frac{R}{N}}. \tag{14}$$

The proof can be found in Appendix D. Note that Lemma 1 indicates that the complexity of the function class is proportional to the width of the neural network $R$ and inversely proportional to the sample size $N$. With Lemma 1, we have the following theorem:

**Theorem 2 (Finite-Sample Identifiability)** *Under the generative model (3), assume that Assumptions 1, 2 and 3 hold. Suppose that $\boldsymbol{x}_\ell$ for $\ell \in [N]$ are i.i.d. samples from $\mathcal{X}$ according to a certain distribution $\mathcal{D}$. Denote any solution of (12) as $\widehat{\boldsymbol{f}} = [\widehat{f}_1, \ldots, \widehat{f}_M]^\top$ and $\widehat{h}_m = \widehat{f}_m \circ g_m$ for $m \in [M]$. Then, with probability of at least $1 - \delta$, the following holds:*

$$\mathbb{E}\left[\left\|\widehat{\boldsymbol{h}}''(\boldsymbol{A}\boldsymbol{s})\right\|_2^2\right] = O\left(\frac{C_d \sqrt{M}\nu}{\sigma_{\min}^2(\boldsymbol{Q}^\top \odot \boldsymbol{B})} + \frac{C_d B^2 C_x \left(\sqrt{R} + \sqrt{2\log(4/\delta)}\right)^{1/2}}{\sigma_{\min}^2(\boldsymbol{Q}^\top \odot \boldsymbol{B})N^{1/4}}\right), \tag{15}$$

*where $\sigma_{\min}^2(\boldsymbol{Q}^\top \odot \boldsymbol{B})$ denotes the smallest singular value of the matrix $\boldsymbol{Q}^\top \odot \boldsymbol{B}$, which is defined in the linear system (11).*

The detailed proof is relegated to Appendix D. The proof consists of three steps: 1) treat the loss function as a supervised regression problem and estimate the "generalization error" using the empirical error and Rademacher complexity; 2) approximate the linear system in (11) using the generalization error and numerical differentiation; and 3) bound and characterize the second-order derivatives based on the approximated linear system.

Theorem 2 implies that the second-order derivatives of $\widehat{h}_m(\cdot)$'s approach zero with sufficiently large sample size $N$. However, the $\nu$ term is not determined by $N$ and it only decreases to zero when $\mathcal{F}$ is a universal function approximator (which could be attained if $R$ is large enough). Therefore, there is always a trade-off between the complexity (encoded by key parameters like depth and width) of the neural network and the effectiveness of model identification given fixed sample size $N$. According to Theorem 2, one hopes to employ an expressive enough neural network for canceling $g_m(\cdot)$, but excessively increasing the expressiveness would hurt the performance under a given $N$—this is consistent with our experience in many neural network learning problems.

## 4 Optimization Design

In this section, we will design an optimization scheme to implement the criterion in (12) Specifically, we propose to the following formulation:

$$\min_{\boldsymbol{Q},\boldsymbol{f}} \; \frac{1}{N}\sum_{\ell=1}^N \left\|\boldsymbol{Q}^\top \boldsymbol{f}(\boldsymbol{x}_\ell)\right\|_2^2 \tag{16a}$$

$$\text{s.t.} \quad \boldsymbol{Q}^\top \boldsymbol{Q} = \boldsymbol{I}, \; f_m(\cdot) \in \mathcal{F}, \; f_m(\cdot) \text{ is invertible.} \tag{16b}$$

Note that the formulation in (16) does not ensure a dense $\boldsymbol{Q}$. Nonetheless, if $\boldsymbol{Q}$ is initialized randomly using any continuous distribution, iterative optimizers would never return a $\boldsymbol{Q}$ that contains zeros, if the iterative algorithm can be approximated by a continuous transformation. We use the neural network structure defined in Assumption 3 to approximate $f_m(\cdot)$. Note that with sufficiently large $R$, the function $f_m(\cdot)$ is a universal approximator [42, 43]. In addition, to promote invertibility, we introduce another neural network $\boldsymbol{r}(\cdot)$, with the same structure as that of $\boldsymbol{f}(\cdot)$, to make sure that $\boldsymbol{f}(\boldsymbol{x}_\ell)$ can be converted back to $\boldsymbol{x}_\ell$. This idea is from autoencoder [44]. The problem is then recast as follows:

$$\min_{\boldsymbol{Q}^\top\boldsymbol{Q}=\boldsymbol{I},\boldsymbol{f},\boldsymbol{r}} \underbrace{\frac{1}{N}\sum_{\ell=1}^N \left\|\boldsymbol{Q}^\top \boldsymbol{f}(\boldsymbol{x}_\ell)\right\|_2^2}_{\mathcal{L}_1} + \lambda \underbrace{\frac{1}{N}\sum_{\ell=1}^N \left\|\boldsymbol{x}_\ell - \boldsymbol{r}(\boldsymbol{f}(\boldsymbol{x}_\ell))\right\|_2^2}_{\mathcal{L}_2} \tag{17}$$

where the hyperparameter $\lambda \geq 0$ is used for balancing the energy between the loss and the regularizer term. In addition, we define $\mathcal{L} = \mathcal{L}_1 + \lambda\mathcal{L}_2$.

The problem can be handled by a block coordinate descent (BCD) algorithm as shown in Algorithm 1. Note that any stochastic optimizer designed for neural network training (e.g., [30]) can be used here for the $\boldsymbol{f}(\cdot)$ and $\boldsymbol{r}(\cdot)$ updates. We use $\boldsymbol{\theta}_f$ and $\boldsymbol{\theta}_r$ to denote the parameters of the corresponding neural networks. Besides, we denote $\widehat{\mathcal{L}}$ and $\widehat{\mathcal{L}}_i$ for $i = 1, 2$ as estimates of $\mathcal{L}$ and $\mathcal{L}_i$ over a random batch $\mathcal{B}$, respectively. For example, $\widehat{\mathcal{L}}_1 = \frac{1}{|\mathcal{B}|}\sum_{\ell\in\mathcal{B}}\|\boldsymbol{Q}^\top\boldsymbol{f}(\boldsymbol{x}_\ell)\|_2^2$, and $\widehat{\mathcal{L}}_2$ is defined in an identical way. The term $\nabla_{\boldsymbol{\theta}_r}\widehat{\mathcal{L}}_2$ denotes the derivative of $\widehat{\mathcal{L}}_2$ taken w.r.t. $\boldsymbol{\theta}_r$. The other derivative terms are defined follwoing the same manner.

We refer to the algorithm as *post-nonlinear subspace identification*. This is because Theorem 1 indicates that the method identifies $\text{range}(\boldsymbol{S}^\top)$ where $\boldsymbol{S} = [\boldsymbol{s}_1, \dots, \boldsymbol{s}_N]$ for $N \geq K$ after removing the nonlinear distortions (technically, this needs $d_m = 0$ for all $m$ (cf. Theorem 1)—which automatically holds if our norm minimization term in the objective becomes zero).

---

**Algorithm 1:** Post-Nonlinear Subspace Identification.

**Data:** $\boldsymbol{x}_\ell$ for $\ell = 1, \cdots, N$
**Result:** $\boldsymbol{f}$
1 **while** *stopping criterion is not reached* **do**
2    $\boldsymbol{Q} \leftarrow \boldsymbol{U}[:, \, M - D : M]$ where $\boldsymbol{U}\boldsymbol{D}\boldsymbol{V}^\top = \text{SVD}(\boldsymbol{F})$ with $\boldsymbol{F} = [\boldsymbol{f}(\boldsymbol{x}_1), \cdots, \boldsymbol{f}(\boldsymbol{x}_N)]$;
3    **while** *stopping criterion is not reached* **do**
4      Draw a random batch $\mathcal{B}$;
5      $\nabla_{\boldsymbol{\theta}_f}\widehat{\mathcal{L}} \leftarrow \nabla_{\boldsymbol{\theta}_f}\widehat{\mathcal{L}}_1 + \lambda\nabla_{\boldsymbol{\theta}_f}\widehat{\mathcal{L}}_2$;
6      $\nabla_{\boldsymbol{\theta}_r}\widehat{\mathcal{L}} \leftarrow \nabla_{\boldsymbol{\theta}_r}\widehat{\mathcal{L}}_2$;
7      Update $\boldsymbol{f}$ and $\boldsymbol{r}$ using $\nabla_{\boldsymbol{\theta}_f}\widehat{\mathcal{L}}$ and $\nabla_{\boldsymbol{\theta}_r}\widehat{\mathcal{L}}$ with any stochastic optimizer, respectively;
8    **end**
9 **end**

---

## 5 Numerical Experiments

### 5.1 Synthetic Data

We generate data following the model in (3). For the neural networks representing $f_m(\cdot)$ and $r_m(\cdot)$, we use $R = 256$ with ReLU activations. We use the Adam optimizer [30] with the initial learning rate being $2e^{-4}$ for the network optimization part. For the hyperparameters, we set $\lambda = 1e^{-4}$. The nonlinear functions $g_m(\cdot)$'s are selected to be variants of $e^x$, $\text{sigmoid}(x)$ and $\tanh(\text{x})$ to guarantee invertibility. The source code can be found online[2].

**Independent Latent Components.** In this simulation, we make $K = 3$ and $M = 5$. Each element of $\boldsymbol{s}_\ell$ is drawn independently from the uniform distribution $U[-1, 1]$ with $N = 10000$, and the mixing matrix $\boldsymbol{A}$ is drawn from the normal distribution. Under this setting, it is obvious that $\boldsymbol{Q} \in \mathbb{R}^{5\times2}$.

Fig. 1 shows the composition $\widehat{h}_m(\cdot) = \widehat{f}_m \circ g_m(\cdot)$. One can see that all the five compositions are visually affine, which means that the algorithm has successfully identified and removed $\boldsymbol{g}(\cdot)$. Prior works on post-nonlinear ICA oftentimes involve computation and sample-size demanding implementations. For example, in [8], a nonparameteric density estimation step of the learned components needs to be involved *in every iteration* in order to measure and promote statistical independence. In comparison, the proposed approach is much more straightforward and easier to implement, as it does not rely on statistical independence.

**Dependent Latent Components.** In this case, we consider the latent components drawn from the probability simplex as in (6). The mixing matrix $\boldsymbol{A}$ is the same as before. Note that the latent components are dependent. Consequently the dimension of the null space is $M - K + 1$ because there are only $\widetilde{K} = K - 1$ free variables in $\boldsymbol{s}_\ell$. For this simulation, we set $N = 10,000$, $K = 3$, and $M = 5$. Accordingly, the dimension of $\mathcal{N}(\boldsymbol{A}^\top)$ is $M - K + 1 = 3$ which makes $\boldsymbol{Q}$ a $5 \times 3$ matrix.

---

[2]https://github.com/llvqi

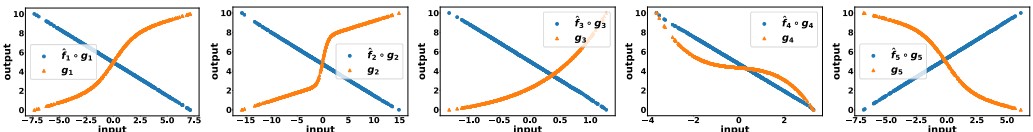

Figure 1: Learned function compositions with independent latent components.

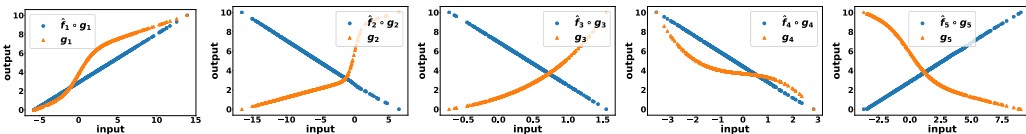

Figure 2: Learned function compositions with dependent latent components.

Fig. 2 shows the results. It is obvious that the nonlinear distortions are successfully removed. Readers might have noticed that in this case the condition in (9) is actually violated since $\widetilde{K}(\widetilde{K}+1)/2 = 3 < 5$. Interestingly, the method still works. This is likely because in this numerical example $\boldsymbol{Q}^\top$ tends to have full Kruskal rank while in our proof we only used that $\mathrm{krank}(\boldsymbol{Q}^\top) = 1$. It turns out that the identifiability condition could actually be much improved if $\boldsymbol{Q}^\top$ has full Kruskal rank. We refer interested readers to further discussions in Appendix A.

**Validating Theorem 2.** In this part, we demonstrate the impact of $R$ (i.e., the width of the employed neural network) on the performance of nonlinearity removal, as stated in Theorem 2. To be specific, we measure the subspace distance (see the definition in [20]) between the estimated $\mathrm{range}(\widehat{\boldsymbol{S}}^\top)$ and the ground truth $\mathrm{range}(\boldsymbol{S}^\top)$, where $\boldsymbol{S} = [\boldsymbol{s}_1, \cdots, \boldsymbol{s}_N]$. This value is within [0,1] with 0 being the best. The setting is the same as that of the independent case.

Table 1 shows the results, which are averaged over 5 random trials. One can see that when $f_m(\cdot)$ is not expressive enough, i.e., when $R = 8$, the performance is far from satisfactory. This is because the term $\nu$ dominates in the bound (15). As the number of neurons increases, the subspace distance keeps decreasing, which indicates that the latent subspace is well identified. However, if the function is overly complex ($R = 1024$), the performance starts to deteriorate since in this case the $R$ becomes dominant on the right hand side of (15). In addition, one can see that larger $N$ leads to a better performance in general, which also validates our claim in (15).

## 5.2 Real Data

**Dataset.** We use the human face electroencephalogram (EEG) dataset[3]. The EEG signals were collected when a subject was shown with pictures of real human faces or scrambled faces. The scrambled face images were generated from the real faces with Fourier transformation by random phase permutations [45]). At every sample, the EEG signal is a 130-dimensional vector $\boldsymbol{x}_\ell$, which are measured through 130 sensors (channels) all over the subject's scalp. The task is to use $\boldsymbol{x}_\ell$ for $\ell = 1, \ldots, N$ as the training data to learn a classifier. The classifier aims to determine whether the subject sees a real face image or scrambled ones when a new $\boldsymbol{x}_\ell$ is collected.

**Metric.** We use a number of unsupervised representation learning (URL) method to extract low-dimensional embeddings of $\boldsymbol{x}_\ell$'s. Then, these embeddings are used to train classifiers based on support vector machines (SVM) and logistic regression (LR). We have $N = 27,692$ samples of $\boldsymbol{x}_\ell$, which are split as training, validation and test sets with 24794, 1449, and 1449 samples, respectively. We use the classification error to serve as an indirect measure of the performance.

We follow the hypothesis in [45]. There, the model is $\boldsymbol{x}_\ell \approx \boldsymbol{g}(\boldsymbol{A}\boldsymbol{s}_\ell)$ with a PNL structure, where $\boldsymbol{s}_\ell$ are the brain generated "clean" electronic signals, and $\boldsymbol{A}$ and $\boldsymbol{g}(\cdot)$ represent the propagation, mixing, and distortions on the sensor end, respectively [23]. Hence, we use our method to remove $\boldsymbol{g}(\cdot)$ and then reduce the dimension of $\boldsymbol{A}\boldsymbol{s}_\ell$ to $K'$ via PCA. Our belief is that if the PNL learning method can remove $\boldsymbol{g}(\cdot)$, then the extracted undistorted signals will benefit training linear classifiers such as SVM

---

[3]https://www.fil.ion.ucl.ac.uk/spm/data/mmfaces/

Table 1: Subspace distance with independent components.

|  | $R = 8$ | $R = 16$ | $R = 32$ | $R = 64$ | $R = 128$ | $R = 256$ | $R = 1024$ |
|---|---|---|---|---|---|---|---|
| $N = 10000$ | 0.47±0.08 | 0.22±0.05 | 0.08±0.02 | 0.05±0.02 | 0.04±0.01 | 0.03±0.01 | 0.07±0.01 |
| $N = 20000$ | 0.44±0.06 | 0.20±0.08 | 0.07±0.02 | 0.05±0.01 | 0.03±0.01 | 0.02±0.01 | 0.06±0.01 |

Table 2: Average classification error±std (best error rate attained in the 5 trials) on the EEG data. ($K'$) is the feature dimension for training the classifier.

|  | Raw | PCA (30) | Autoencoder (50) | Proposed (30) | nICA [16] (30) | PNL-ICA [9] (30) |
|---|---|---|---|---|---|---|
| SVM | 0.46±0.05 (0.43) | 0.43±0.02 (0.41) | 0.43±0.03 (0.40) | **0.40±0.03 (0.35)** | 0.44±0.02 (0.43) | 0.46±0.05 (0.38) |
| LR | 0.47±0.04 (0.42) | 0.45±0.04 (0.41) | 0.46±0.03 (0.41) | **0.39±0.03 (0.35)** | 0.43±0.03 (0.39) | 0.44±0.03 (0.39) |

and LR. We include PCA (applied onto the raw data), autoencoder, and nonlinear ICA (nICA) based approaches [9, 16] as baselines.

**Results.** Table 2 shows the classification error. The results are averaged over 5 random trials. For all methods, the reduced dimension $K'$ (for our method, we have $D = M - K'$) is selected over $\{10, 30, 50, 70\}$ using the validation set. We use fully connected networks that have two hidden layers and 256 neurons on each layer. The same network structure is used for the autoencoder baseline. The activation functions are all ReLU. We use Adam [30] optimizer for neural network updates, where the initial learning rate is $1e^{-4}$.

One can see that our method has the lowest classification error, compared to other baselines. Note that brain signal classification problem is quite challenging, and thus using the raw data only outputs an accuracy that is slightly better than random guesses. Using our method, the average classification errors are reduced by 6% and 7% from the those using the raw data for SVM and LR, respectively, which is substantial. This also shows the usefulness of the PNL model for EEG data. The PCA is a linear method that does not capture the nonlinearity of the data, and thus the result under PCA is less promising. The autoencoder can be understood as a nonlinearity removal method, but it does not ensure provable $g(\cdot)$ removal. The nICA variants also exhibit higher error rates relative to the proposed method. This may be because the ICA frameworks hinge on structural assumptions on the latent components, which may not always be available in practice.

## 6 Conclusion

In this work, we revisited the PNL model from a model identification perspective. We proposed a new framework to identify and remove the unknown nonlinear distortions in the data generating/acquisition process. Compared to existing works, the proposed approach does not require the latent components to satisfy specific structural assumptions, e.g., statistical independence. Consequently, our new result offers a PNL model identification tool to a much wider spectrum of applications. In addition, we provided analysis of the cases where only finite samples and non-universal function learners are available. We designed a constrained optimization scheme to implement the proposed learning criterion. Our theoretical claims were supported by experiment results.

There are a number of limitations. First, the PNL model would incur high computational cost when the data feature dimension $M$ is large, since each dimension induces an individual neural network. Second, there are meaningful applications where the $A$ matrix may not have a nontrivial orthogonal complement (e.g., in PNL-based causal direction estimation [12]), yet this current framework does not guarantee nonlinearity removal in those cases. Third, the PNL problem's learning criterion is a nonconvex optimization problem. The proposed algorithm does not guarantee to solve the formulated learning criterion, which creates a gap between the identifiable results (by assuming optimization optimality) and the attainable results in practice.

**Acknowledgement**: This work was supported by the National Science Foundation (NSF) CAREER Award under Project ECCS-2144889.

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
