# Supplementary Material

## A When Does Exact Equivalence Between (16) and (5) Hold?

It is interesting to discuss when using the formulation in (16) can exactly approximate (5). To answer this question, it is worth noting that if $\boldsymbol{Q}$ is a random matrix drawn from any jointly continuous distribution, with $\boldsymbol{Q}^\top \boldsymbol{Q} = \boldsymbol{I}$, then, $\|\boldsymbol{Q}\|_0 = MD$ holds with probability one—as the probability of any continuous random variable being zero is zero.

Note that according to (16), the optimal solution of $\boldsymbol{Q}$ is given by

$$\boldsymbol{Q} \leftarrow \underset{\boldsymbol{Q}^\top \boldsymbol{Q} = \boldsymbol{I}}{\arg\min} \, \mathbb{E}\left[ \left\| \boldsymbol{Q}^\top \boldsymbol{f}(\boldsymbol{x}) \right\|_2^2 \right]$$
$$\Longleftrightarrow \quad \boldsymbol{Q} \leftarrow \underset{\boldsymbol{Q}^\top \boldsymbol{Q} = \boldsymbol{I}}{\arg\min} \, \mathrm{Tr}\left( \boldsymbol{Q}^\top \mathbb{E}\left[ \boldsymbol{f}(\boldsymbol{x}) \boldsymbol{f}(\boldsymbol{x})^\top \right] \boldsymbol{Q} \right).$$

The solution $\boldsymbol{Q}$ of the above consists of the eigenvectors of the covariance matrix of $\boldsymbol{f}(\boldsymbol{x})$ corresponding to the $D$ smallest eigenvalues. Assume that $p(\boldsymbol{x})$ is a continuous joint PDF, and that $\boldsymbol{f}(\cdot)$ is continuous and invertible. Then, $\boldsymbol{Q}$ is a continuous function of $\mathbb{E}\left[ \boldsymbol{f}(\boldsymbol{x}) \boldsymbol{f}(\boldsymbol{x})^\top \right]$ (which is also a continuous function of $\boldsymbol{x}$ by composition). It is known that, when $\mathbb{E}\left[ \boldsymbol{f}(\boldsymbol{x}) \boldsymbol{f}(\boldsymbol{x})^\top \right]$'s eigenvalues are all distinct, then $\boldsymbol{Q}$ is a continuous function of $\mathbb{E}\left[ \boldsymbol{f}(\boldsymbol{x}) \boldsymbol{f}(\boldsymbol{x})^\top \right]$ [46], thereby a continuous function of $\boldsymbol{x}$—which means $\boldsymbol{Q}$ is a continuous random matrix. A special case is that when $\boldsymbol{f}(\boldsymbol{x})$ is a normal random vector, the columns of $\boldsymbol{Q}$ are uniformly distributed on a unit ball [47]. We should remark that some conditions mentioned above (e.g., distinct eigenvalues) for the exact equivalence of (16) and (5) are not easy to check or meet. Nonetheless, these are only sufficient conditions—violating them does not mean a dense $\boldsymbol{Q}$ cannot be attained. As we mentioned, our approximation in (16) for (5) works quite well—and dense $\boldsymbol{Q}$'s were always observed in our experiments.

## B Full Kruskal Rank of The Left Matrix in (11) of Theorem 1

In this section, we show that the matrix $\boldsymbol{B}$ as defined in (11) has full Kruskal rank with probability one in different cases.

First, consider the case that $\boldsymbol{B}$ is a tall matrix, i.e., $\frac{\widetilde{K}(\widetilde{K}+1)}{2} \geq M$. We only need to show that the $\boldsymbol{B}$ matrix is full column rank. This can be obtained by showing that there exists a particular case such that an $M \times M$ submatrix of $\boldsymbol{B}$ has full column rank. The reason is that the determinant of any $M \times M$ submatrix of $\boldsymbol{B}$ is a polynomial of $\boldsymbol{A}$, and a polynomial is nonzero almost everywhere if it is nonzero somewhere [48, Lemma 2].

Consider a special case where $\boldsymbol{A}$ is a Vandermonde matrix, i.e., $\boldsymbol{a}_k = [1, z_k, z_k^2, \ldots, z_k^{M-1}]^\top$, and $z_i \neq z_j$. Hence, the corresponding $\boldsymbol{B}$ has the following form

$$\begin{bmatrix} 1 & z_1^2 & \ldots & z_1^{2(M-1)} \\ \ldots & \ldots & & \ldots \\ 1 & z_K^2 & \ldots & z_K^{2(M-1)} \\ 1 & z_1 z_2 & \ldots & (z_1 z_2)^{M-1} \\ \ldots & \ldots & & \ldots \\ 1 & z_{K-1} z_K & \ldots & (z_{K-1} z_K)^{M-1} \end{bmatrix} \tag{18}$$

Note that one can always construct such a sequence—e.g., $z_1 = 1, z_2 = 1.1, z_3 = 1.11, \ldots$ such that any $M$ rows of the matrix in (18) is full rank. This means that the linear combination of this second order homogeneous polynomials is not identically zero, which implies that it is non-zero almost everywhere; see a similar argument in [19, 20]. Thus, it further implies that the matrix $\boldsymbol{B}$ has a Kruskal rank of $M$ almost surely when $\frac{\widetilde{K}(\widetilde{K}+1)}{2} \geq M$.

On the other hand, when $\frac{\widetilde{K}(\widetilde{K}+1)}{2} < M$, for any $\frac{\widetilde{K}(\widetilde{K}+1)}{2}$ columns we have the following form

$$
\underbrace{\begin{bmatrix}
a_{1,1}^2 & \cdots & a_{1,\frac{\widetilde{K}(\widetilde{K}+1)}{2}}^2 \\
\cdots & \vdots & \cdots \\
a_{\widetilde{K},1}^2 & \cdots & a_{\widetilde{K},\frac{\widetilde{K}(\widetilde{K}+1)}{2}}^2 \\
a_{1,1}a_{2,1} & \cdots & a_{1,\frac{\widetilde{K}(\widetilde{K}+1)}{2}}a_{2,\frac{\widetilde{K}(\widetilde{K}+1)}{2}} \\
\cdots & \vdots & \cdots \\
a_{\widetilde{K}-1,1}a_{\widetilde{K},1} & \cdots & a_{\widetilde{K}-1,\frac{\widetilde{K}(\widetilde{K}+1)}{2}}a_{\widetilde{K},\frac{\widetilde{K}(\widetilde{K}+1)}{2}}
\end{bmatrix}}_{\frac{\widetilde{K}(\widetilde{K}+1)}{2}}
$$

where the columns are reordered from 1 to $\frac{\widetilde{K}(\widetilde{K}+1)}{2}$. Then, the same proof technique may apply here. Thus, by combining the two cases, it is clear that $B$ has the following Kruskal rank with probability one:

$$
\min\left(\frac{\widetilde{K}(\widetilde{K}+1)}{2}, M\right).
$$

## C Proof of Lemma 1

By the definition of $\mathcal{F}$ in Assumption 3, we have

$$
\begin{aligned}
|f_m(x) - f_m(0)| &= |\boldsymbol{w}_2^\top \boldsymbol{\zeta}(\boldsymbol{w}_1 x) - \boldsymbol{w}_2^\top \boldsymbol{\zeta}(\boldsymbol{0})| \\
&\leq \|\boldsymbol{w}_2\|_2 \|\boldsymbol{\zeta}(\boldsymbol{w}_1 x) - \boldsymbol{\zeta}(\boldsymbol{0})\|_2 \\
&\leq B\|\boldsymbol{w}_1 x - \boldsymbol{0}\|_2 \\
&\leq B^2 C_x, \\
\implies |f_m(x)| &\leq B^2 C_x,
\end{aligned}
$$

where the last inequality is because $f_m(0) = 0$. Besides, the Rademacher complexity of the function class $\mathcal{F}$ is bounded by [49]

$$
\mathfrak{R}(\mathcal{F}) \leq 2B^2 C_x \sqrt{\frac{R}{N}}.
$$

The invertible function class subset is also upper bounded by this complexity. Note that the function class that we are interested in is

$$
\left\|\boldsymbol{Q}^\top \boldsymbol{f}(\boldsymbol{x}_\ell)\right\|_2^2 \in \mathcal{T}: \ \mathbb{R}^M \to \mathbb{R},
$$

which has the following Rademacher complexity

$$
\begin{aligned}
\mathfrak{R}(\mathcal{T}) &= \frac{1}{N}\mathbb{E}_{\boldsymbol{\sigma}}\left[\sup_{f_1(\cdot),\cdots,f_M(\cdot)\in\mathcal{F}}\sum_{\ell=1}^{N}\sigma_\ell\left\|\boldsymbol{Q}^\top\boldsymbol{f}(\boldsymbol{x}_\ell)\right\|_2^2\right] \\
&= \frac{1}{N}\mathbb{E}_{\boldsymbol{\sigma}}\left[\sup_{f_1(\cdot),\cdots,f_M(\cdot)\in\mathcal{F}}\sum_{\ell=1}^{N}\sum_{j=1}^{D}\sigma_\ell\left(\boldsymbol{q}_j^\top\boldsymbol{f}(\boldsymbol{x}_\ell)\right)^2\right] \\
&\leq \frac{1}{N}\mathbb{E}_{\boldsymbol{\sigma}}\left[\sup_{f_1(\cdot),\cdots,f_M(\cdot)\in\mathcal{F}}\sum_{\ell=1}^{N}\sum_{j=1}^{D}\sigma_\ell\|\boldsymbol{q}_j\|_2^2\|\boldsymbol{f}(\boldsymbol{x}_\ell)\|_2^2\right] \\
&= \frac{D}{N}\mathbb{E}_{\boldsymbol{\sigma}}\left[\sup_{f_1(\cdot),\cdots,f_M(\cdot)\in\mathcal{F}}\sum_{\ell=1}^{N}\sigma_\ell\|\boldsymbol{f}(\boldsymbol{x}_\ell)\|_2^2\right]\ (\text{ since } \|\boldsymbol{q}_j\|_2^2=1) \\
&= \frac{D}{N}\mathbb{E}_{\boldsymbol{\sigma}}\left[\sup_{f_1(\cdot),\cdots,f_M(\cdot)\in\mathcal{F}}\sum_{\ell=1}^{N}\sum_{m=1}^{M}\sigma_\ell\,|f_m(x_{\ell,m})|^2\right] \\
&\leq 2DMB^2C_x\mathcal{R}(\mathcal{F}) \\
&= 2DMB^4C_x^2\sqrt{\frac{R}{N}},
\end{aligned}
$$

where the last inequality is because of the Lipschitz composition property of the Rademacher complexity [50]. Besides, we also have $\left\|\boldsymbol{Q}^\top\boldsymbol{f}(\boldsymbol{x}_\ell)\right\|_2^2\leq DB^4C_x^2$.

# D  Proof of Theorem 2

Our proof uses the proof technique from [19, 28, 29] that considered the simplex-structured and multiview nonlinear models, with nontrivial modifications to accommodate our generative model.

To perform finite-sample analysis, consider the finite-sample version formulation

$$
\min_{\boldsymbol{Q},\boldsymbol{f}}\ \frac{1}{N}\sum_{\ell=1}^{N}\left\|\boldsymbol{Q}^\top\boldsymbol{f}(\boldsymbol{x}_\ell)\right\|_2^2 \tag{19a}
$$

$$
\text{s.t.}\ \ \|\boldsymbol{Q}\|_0=MD,\ \boldsymbol{Q}^\top\boldsymbol{Q}=\boldsymbol{I},\ f_m(\cdot)\in\mathcal{F}, \tag{19b}
$$

where $\boldsymbol{Q}\in\mathbb{R}^{M\times D}$, $f_m$'s are the nonlinear functions that we aim to learn.

The overall idea of proof is to use the empirical error on (19a) to bound the true error. Then use the numerical differentiation to estimate the second-order derivatives of $h''_m$. The framework is similar to that of [19,28].

The problem can be regarded as a regression problem with data tuples $\{\boldsymbol{x}_\ell, \boldsymbol{0}_D\}_{\ell=1}^N$. According to [50,51], we have

$$
\begin{aligned}
\mathbb{E}\left[\left\|\boldsymbol{Q}^\top \boldsymbol{f}(\boldsymbol{x}_\ell)\right\|_2^2\right] &\leq \frac{1}{N}\sum_{\ell=1}^N \left\|\boldsymbol{Q}^\top \boldsymbol{f}(\boldsymbol{x}_\ell)\right\|_2^2 + 2\Re + 4DMB^4C_x^2\sqrt{\frac{2\log(4/\delta)}{N}} \\
&= \frac{1}{N}\sum_{\ell=1}^N \left\|\boldsymbol{Q}^\top (\boldsymbol{f}(\boldsymbol{x}_\ell) - \boldsymbol{u}(\boldsymbol{x}_\ell) + \boldsymbol{u}(\boldsymbol{x}_\ell))\right\|_2^2 + 2\Re + 4DMB^4C_x^2\sqrt{\frac{2\log(4/\delta)}{N}} \\
&\overset{(a)}{\leq} \frac{1}{N}\sum_{\ell=1}^N \left(\left\|\boldsymbol{Q}^\top (\boldsymbol{f}(\boldsymbol{x}_\ell) - \boldsymbol{u}(\boldsymbol{x}_\ell))\right\|_2 + \left\|\boldsymbol{Q}^\top \boldsymbol{u}(\boldsymbol{x}_\ell)\right\|_2\right)^2 + 2\Re + 4DMB^4C_x^2\sqrt{\frac{2\log(4/\delta)}{N}} \\
&= \frac{1}{N}\sum_{\ell=1}^N \left\|\boldsymbol{Q}^\top (\boldsymbol{f}(\boldsymbol{x}_\ell) - \boldsymbol{u}(\boldsymbol{x}_\ell))\right\|_2^2 + 2\Re + 4DMB^4C_x^2\sqrt{\frac{2\log(4/\delta)}{N}} \\
&\overset{(b)}{\leq} DM\nu^2 + 2\Re + 4DMB^4C_x^2\sqrt{\frac{2\log(4/\delta)}{N}} \\
&\leq DM\nu^2 + 4DMB^4C_x^2\sqrt{\frac{R}{N}} + 4DMB^4C_x^2\sqrt{\frac{2\log(4/\delta)}{N}},
\end{aligned}
$$

since $\left\|\boldsymbol{Q}^\top \boldsymbol{u}(\boldsymbol{x}_\ell)\right\|_2 = 0$ holds for all $\ell$, where (a) is by triangle inequality and (b) is by both triangle inequality and Assumption 1.

Next, we estimate the second order derivative given the solution of (12). Suppose for any sample $\boldsymbol{x}_\ell$ we have

$$
\left\|\boldsymbol{Q}^\top \boldsymbol{f}(\boldsymbol{x}_\ell)\right\|_2^2 = \epsilon_\ell
$$

where $\varepsilon_\ell \geq 0$ such that $\mathbb{E}\left[\varepsilon_\ell\right] \leq \varepsilon$.

Consider each column of $\boldsymbol{Q}$ separately. For each column $\boldsymbol{q}_k$ with $k = 1, \cdots, D$, we have

$$
\phi_k(\boldsymbol{s}_\ell) := \boldsymbol{q}_k^\top \widehat{\boldsymbol{h}}(\boldsymbol{A}\boldsymbol{s}_\ell) = \pm\sqrt{\epsilon_{\ell,k}} \tag{20}
$$

where $\varepsilon_\ell = \sum_{k=1}^D \varepsilon_{\ell,k}$ with each $\varepsilon_{\ell,k} \geq 0$. In the following part, we will estimate the second-order derivative $\frac{\partial^2 \phi_k(\boldsymbol{s})}{\partial s_i^2}$ and the cross second-order derivative $\frac{\partial^2 \phi_k(\boldsymbol{s})}{\partial s_i s_j}$, respectively.

## D.1 Estimating the Second-Order Derivatives

Define $\Delta\boldsymbol{s}_i = [0,\ldots,\Delta s_i,\ldots,0]^\top$ for $i = 1,\ldots,\widetilde{K}$, and $\boldsymbol{s}_{\widehat{\ell}} = \boldsymbol{s}_\ell + \Delta\boldsymbol{s}_i$ and $\boldsymbol{s}_{\widetilde{\ell}} = \boldsymbol{s}_\ell - \Delta\boldsymbol{s}_i$. Therefore, we have

$$
\begin{aligned}
\phi_k(\boldsymbol{s}_{\widehat{\ell}}) &= \boldsymbol{q}_k^\top \widehat{\boldsymbol{h}}(\boldsymbol{A}(\boldsymbol{s}_\ell + \Delta\boldsymbol{s}_i)) = \pm\sqrt{\varepsilon_{\widehat{\ell},k}}, \\
\phi_k(\boldsymbol{s}_{\widetilde{\ell}}) &= \boldsymbol{q}_k^\top \widehat{\boldsymbol{h}}(\boldsymbol{A}(\boldsymbol{s}_\ell - \Delta\boldsymbol{s}_i)) = \pm\sqrt{\varepsilon_{\widetilde{\ell},k}},
\end{aligned} \tag{21}
$$

where $\mathbb{E}[\varepsilon_{\widehat{\ell},k}] = \mathbb{E}[\varepsilon_{\widetilde{\ell},k}] \leq \frac{\varepsilon}{D}$.

For any continuous function $\omega(z)$ that admits non-vanishing 4th order derivatives, the second order derivative at $z$ can be estimated as follows [52]

$$
\omega''(z) = \frac{\omega(z + \Delta z) - 2\omega(z) + \omega(z - \Delta z)}{\Delta z^2} - \frac{\Delta z^2}{12}\omega^{(4)}(\xi),
$$

where $\xi \in (z - \Delta z, z + \Delta z)$.

Following this definition, one can see that

$$
\frac{\partial^2 \phi_k(\boldsymbol{s})}{\partial s_i^2} = \frac{\pm\sqrt{\varepsilon_{\widehat{\ell},k}} \mp 2\sqrt{\varepsilon_{\ell,k}} \pm \sqrt{\varepsilon_{\widetilde{\ell},k}}}{\Delta s_i^2} - \frac{\Delta s_i^2}{12}\phi^{(4)}(\boldsymbol{\xi}_i),
$$

where $\boldsymbol{\xi}_i \in (\boldsymbol{s}_{\widetilde{\ell}}, \boldsymbol{s}_{\widehat{\ell}})$. Consequently, we have the following inequalities

$$\left| \sum_{m=1}^{M} q_{m,k} a_{m,i}^2 \widehat{h}_m''(\boldsymbol{A}\boldsymbol{s}_\ell) \right| = \left| \frac{\pm\sqrt{\varepsilon_{\widehat{\ell},k}} \mp 2\sqrt{\varepsilon_{\ell,k}} \pm \sqrt{\varepsilon_{\widetilde{\ell},k}}}{\Delta s_i^2} - \frac{\Delta s_i^2}{12} \phi^{(4)}(\boldsymbol{\xi}_i) \right|$$

$$\leq \frac{\sqrt{\varepsilon_{\widehat{\ell},k}} + 2\sqrt{\varepsilon_{\ell,k}} + \sqrt{\varepsilon_{\widetilde{\ell},k}}}{\Delta s_i^2} + \frac{\Delta s_i^2}{12} \left| \phi^{(4)}(\boldsymbol{\xi}_i) \right|.$$

By taking expectation, we have the following holds with probability at least $1 - \delta$

$$\mathbb{E}\left[ \left| \sum_{m=1}^{M} q_{m,k} a_{m,i}^2 \widehat{h}_m''(\boldsymbol{A}\boldsymbol{s}_\ell) \right| \right] \leq \frac{\mathbb{E}\left[ \sqrt{\varepsilon_{\widehat{\ell},k}} \right] + 2\mathbb{E}\left[ \sqrt{\varepsilon_{\ell,k}} \right] + \mathbb{E}\left[ \sqrt{\varepsilon_{\widetilde{\ell},k}} \right]}{\Delta s_i^2} + \frac{\Delta s_i^2}{12} \left| \phi^{(4)}(\boldsymbol{\xi}_i) \right|$$

$$\leq \frac{4\sqrt{\varepsilon}}{\sqrt{D}\Delta s_i^2} + \frac{\left| \phi^{(4)}(\boldsymbol{\xi}_i) \right| \Delta s_i^2}{12}, \tag{22}$$

where the second inequality is by the Jensen's inequality

$$\mathbb{E}\left[ \sqrt{\varepsilon_{\ell,k}} \right] \leq \sqrt{\mathbb{E}\left[ \varepsilon_{\ell,k} \right]} \leq \sqrt{\frac{\varepsilon}{D}},$$

which holds by the concavity of $\sqrt{x}$ when $x \geq 0$.

We are interested in finding the best upper bound, i.e.,

$$\inf_{\Delta s_i} \frac{4\sqrt{\varepsilon}}{\sqrt{D}\Delta s_i^2} + \frac{\left| \phi^{(4)}(\boldsymbol{\xi}) \right|}{12} \Delta s_i^2. \tag{23}$$

This is an convex problem with $\Delta s_i \in (0, \infty)$ which can be solved to global optimal. By taking derivative w.r.t. $\Delta s_i$, we have the minimizer

$$\Delta s_i = \left( \frac{48\sqrt{\varepsilon}}{\sqrt{D} \left| \phi^{(4)}(\boldsymbol{\xi}_i) \right|} \right)^{1/4},$$

which gives the optimal

$$\frac{2\sqrt{3 \left| \phi^{(4)}(\boldsymbol{\xi}_i) \right|} \varepsilon^{1/4}}{3 D^{1/4}}.$$

Thus we have

$$\left| \mathbb{E}\left[ \sum_{m=1}^{M} q_{m,j} a_{m,i}^2 \widehat{h}_m''(\boldsymbol{A}\boldsymbol{s}_\ell) \right] \right| \leq \frac{2\sqrt{3 \left| \phi^{(4)}(\boldsymbol{\xi}_i) \right|} \varepsilon^{1/4}}{3 D^{1/4}}.$$

Given that $N$ is fixed, one may pick $\varepsilon = DM\nu^2 + 4DMB^4 C_x^2 \sqrt{\frac{R}{N}} + 4DMB^4 C_x^2 \sqrt{\frac{2\log(4/\delta)}{N}}$, which gives us

$$\left| \mathbb{E}\left[ \sum_{m=1}^{M} q_{m,k} a_{m,i}^2 \widehat{h}_m''(\boldsymbol{A}\boldsymbol{s}_\ell) \right] \right| \leq \frac{2\sqrt{3C_d} \left( DM\nu^2 + 4DMB^4 C_x^2 \sqrt{\frac{R}{N}} + 4DMB^4 C_x^2 \sqrt{\frac{2\log(4/\delta)}{N}} \right)^{1/4}}{3 D^{1/4}}$$

$$\leq \frac{2\sqrt{3C_d}}{3} \left( M\nu^2 + 4MB^4 C_x^2 \sqrt{\frac{R}{N}} + 4MB^4 C_x^2 \sqrt{\frac{2\log(4/\delta)}{N}} \right)^{1/4}. \tag{24}$$

## D.2 Estimating the Cross Derivatives

To show the bound for cross-derivatives, we define

$$\Delta s_{ij}^{++} = [\mathbf{0}, \ldots, +\Delta s_i, \ldots, \mathbf{0}, \ldots, +\Delta s_j, \ldots, \mathbf{0}]^\top,$$
$$\Delta s_{ij}^{+-} = [\mathbf{0}, \ldots, +\Delta s_i, \ldots, \mathbf{0}, \ldots, -\Delta s_j, \ldots, \mathbf{0}]^\top,$$
$$\Delta s_{ij}^{-+} = [\mathbf{0}, \ldots, -\Delta s_i, \ldots, \mathbf{0}, \ldots, +\Delta s_j, \ldots, \mathbf{0}]^\top,$$
$$\Delta s_{ij}^{--} = [\mathbf{0}, \ldots, -\Delta s_i, \ldots, \mathbf{0}, \ldots, -\Delta s_j, \ldots, \mathbf{0}]^\top,$$

with $\Delta s_i > 0$ and $\Delta s_j > 0$ for any $i, j \in [K]$ with $i < j$.

Define $s_{\widehat{\ell}} = s_\ell + \Delta s_{ij}^{++}$, $s_{\widetilde{\ell}} = s_\ell + \Delta s_{ij}^{+-}$, $s_{\overline{\ell}} = s_\ell + \Delta s_{ij}^{-+}$, and $s_{\ell'} = s_\ell + \Delta s_{ij}^{--}$. Then, similarly we have

$$
\begin{aligned}
\phi_k(s_{\widehat{\ell}}) &= q_k^\top \widehat{h}(A(s_\ell + \Delta s_{ij}^{++})) = \sqrt{\varepsilon_{\widehat{\ell},k}}, \\
\phi_k(s_{\widetilde{\ell}}) &= q_k^\top \widehat{h}(A(s_\ell + \Delta s_{ij}^{+-})) = \sqrt{\varepsilon_{\widetilde{\ell},k}}, \\
\phi_k(s_{\overline{\ell}}) &= q_k^\top \widehat{h}(A(s_\ell + \Delta s_{ij}^{-+})) = \sqrt{\varepsilon_{\overline{\ell},k}}, \\
\phi_k(s_{\ell'}) &= q_k^\top \widehat{h}(A(s_\ell + \Delta s_{ij}^{--})) = \sqrt{\varepsilon_{\ell',k}}.
\end{aligned}
\tag{25}
$$

where $\mathbb{E}[\varepsilon_{\ell,k}] \leq \frac{\varepsilon}{D}$ for any $\ell$.

By Lemma 6 in [19], the cross derivative of a continuous function $\psi(x, y)$ is estimated as

$$
\begin{aligned}
\frac{\partial^2 \psi(x,y)}{\partial x \partial y} &= \frac{\psi(x+\Delta x, y+\Delta y) - \psi(x+\Delta x, y-\Delta y)}{4\Delta x \Delta y} - \frac{\psi(x-\Delta x, y+\Delta y) - \psi(x-\Delta x, y-\Delta y)}{4\Delta x \Delta y} \\
&\quad - \frac{\Delta x^2}{6} \frac{\partial^4 \psi(\xi_{11}, \xi_{21})}{\partial x^3 \partial y} - \frac{\Delta y^2}{6} \frac{\partial^4 \psi(\xi_{12}, \xi_{22})}{\partial x \partial y^3} - \frac{\Delta x^3}{48 \Delta y} \left( \frac{\partial^4 \psi(\xi_{13}, \xi_{23})}{\partial x^4} - \frac{\partial^4 \psi(\xi_{14}, \xi_{24})}{\partial x^4} \right) \\
&\quad - \frac{\Delta x \Delta y}{8} \left( \frac{\partial^4 \psi(\xi_{15}, \xi_{25})}{\partial x^2 \partial y^2} - \frac{\partial^4 \psi(\xi_{16}, \xi_{26})}{\partial x^2 \partial y^2} \right) - \frac{\Delta y^3}{48 \Delta x} \left( \frac{\partial^4 \psi(\xi_{17}, \xi_{27})}{\partial y^4} - \frac{\partial^4 \psi(\xi_{18}, \xi_{28})}{\partial y^4} \right),
\end{aligned}
$$

where $\xi_{1i} \in (x - \Delta x, x + \Delta x)$ and $\xi_{2i} \in (y - \Delta y, y + \Delta y)$ for $i \in \{1, \cdots, 8\}$. Then, we have

$$
\begin{aligned}
\frac{\partial^2 \phi_k(s)}{\partial s_i \partial s_j} &= \frac{\pm\sqrt{\varepsilon_{\widehat{\ell},k}} \mp \sqrt{\varepsilon_{\widetilde{\ell},k}} \mp \sqrt{\varepsilon_{\overline{\ell},k}} \pm \sqrt{\varepsilon_{\ell',k}}}{4\Delta s_i \Delta s_j} - \frac{\Delta s_i^2}{6} \frac{\partial^4 \phi(\xi_{ij}^{(1)})}{\partial s_i^3 \partial s_j} - \frac{\Delta s_j^2}{6} \frac{\partial^4 \phi(\xi_{ij}^{(2)})}{\partial s_i \partial s_j^3} \\
&\quad - \frac{\Delta s_i^3}{48 \Delta s_j} \left( \frac{\partial^4 \phi(\xi_{ij}^{(3)})}{\partial s_i^4} - \frac{\partial^4 \phi(\xi_{ij}^{(4)})}{\partial s_i^4} \right) - \frac{\Delta s_i \Delta s_j}{8} \left( \frac{\partial^4 \phi(\xi_{ij}^{(5)})}{\partial s_i^2 \partial s_j^2} - \frac{\partial^4 \phi(\xi_{ij}^{(6)})}{\partial s_i^2 \partial s_j^2} \right) \\
&\quad - \frac{\Delta s_j^3}{48 \Delta s_i} \left( \frac{\partial^4 \phi(\xi_{ij}^{(7)})}{\partial s_j^4} - \frac{\partial^4 \phi(\xi_{ij}^{(8)})}{\partial s_j^4} \right),
\end{aligned}
$$

where $\xi_{ij}^{(t)}$ satisfies

$$\xi_{ij}^{(t)} = \theta^{(t)} s_{\widehat{\ell}} + (1 - \theta^{(t)}) s_{\ell'}, \, t \in \{1, \cdots, 8\},$$

with $\theta^{(t)} \in (0, 1)$, is a vector such that $[\xi_{ij}^{(t)}]_i \in (s_{i,\ell} - \Delta s_i, s_{i,\ell} + \Delta s_i)$ and $[\xi_{ij}^{(t)}]_j \in (s_{j,\ell} - \Delta s_j, s_{j,\ell} + \Delta s_j)$. Consequently, we have the following

$$
\begin{aligned}
\left| \frac{\partial^2 \phi_k(s)}{\partial s_i \partial s_j} \right| &\leq \frac{\sqrt{\varepsilon_{\widehat{\ell},k}} + \sqrt{\varepsilon_{\widetilde{\ell},k}} + \sqrt{\varepsilon_{\overline{\ell},k}} + \sqrt{\varepsilon_{\ell',k}}}{4\Delta s_i \Delta s_j} + \frac{\Delta s_i^2}{6} \left| \frac{\partial^4 \phi(\xi_{ij}^{(1)})}{\partial s_i^3 \partial s_j} \right| + \frac{\Delta s_j^2}{6} \left| \frac{\partial^4 \phi(\xi_{ij}^{(2)})}{\partial s_i \partial s_j^3} \right| \\
&\quad + \frac{\Delta s_i^3}{48 \Delta s_j} \left( \left| \frac{\partial^4 \phi(\xi_{ij}^{(3)})}{\partial s_i^4} \right| + \left| \frac{\partial^4 \phi(\xi_{ij}^{(4)})}{\partial s_i^4} \right| \right) + \frac{\Delta s_i \Delta s_j}{8} \left( \left| \frac{\partial^4 \phi(\xi_{ij}^{(5)})}{\partial s_i^2 \partial s_j^2} \right| + \left| \frac{\partial^4 \phi(\xi_{ij}^{(6)})}{\partial s_i^2 \partial s_j^2} \right| \right) \\
&\quad + \frac{\Delta s_j^3}{48 \Delta s_i} \left( \left| \frac{\partial^4 \phi(\xi_{ij}^{(7)})}{\partial s_j^4} \right| + \left| \frac{\partial^4 \phi(\xi_{ij}^{(8)})}{\partial s_j^4} \right| \right).
\end{aligned}
$$

Taking expectation and by Jensen's inequality, we have

$$\left| \mathbb{E}\left[ \sum_{m=1}^{M} q_{m,k} a_{m,i} a_{m,j} \widehat{h}_m''(\boldsymbol{A}\boldsymbol{s}_\ell) \right] \right| \leq \frac{\sqrt{\varepsilon}}{\sqrt{D}\Delta s_i \Delta s_j} + \frac{\Delta s_i^2}{6}\left| \frac{\partial^4 \phi(\boldsymbol{\xi}_{ij}^{(1)})}{\partial s_i^3 \partial s_j} \right| + \frac{\Delta s_j^2}{6}\left| \frac{\partial^4 \phi(\boldsymbol{\xi}_{ij}^{(2)})}{\partial s_i \partial s_j^3} \right|$$
$$+ \frac{\Delta s_i^3}{48\Delta s_j}\left( \left| \frac{\partial^4 \phi(\boldsymbol{\xi}_{ij}^{(3)})}{\partial s_i^4} \right| + \left| \frac{\partial^4 \phi(\boldsymbol{\xi}_{ij}^{(4)})}{\partial s_i^4} \right| \right) + \frac{\Delta s_i \Delta s_j}{8}\left( \left| \frac{\partial^4 \phi(\boldsymbol{\xi}_{ij}^{(5)})}{\partial s_i^2 \partial s_j^2} \right| + \left| \frac{\partial^4 \phi(\boldsymbol{\xi}_{ij}^{(6)})}{\partial s_i^2 \partial s_j^2} \right| \right)$$
$$+ \frac{\Delta s_j^3}{48\Delta s_i}\left( \left| \frac{\partial^4 \phi(\boldsymbol{\xi}_{ij}^{(7)})}{\partial s_j^4} \right| + \left| \frac{\partial^4 \phi(\boldsymbol{\xi}_{ij}^{(8)})}{\partial s_j^4} \right| \right).$$

Note that we aim to find the smallest upper bound, i.e.,

$$\inf_{\Delta s_i, \Delta s_j} \frac{\sqrt{\varepsilon_0}}{\sqrt{D}\Delta s_i \Delta s_j} + \frac{\Delta s_i^2}{6}\left| \frac{\partial^4 \phi(\boldsymbol{\xi}_{ij}^{(1)})}{\partial s_i^3 \partial s_j} \right| + \frac{\Delta s_j^2}{6}\left| \frac{\partial^4 \phi(\boldsymbol{\xi}_{ij}^{(2)})}{\partial s_i \partial s_j^3} \right| + \frac{\Delta s_i^3}{48\Delta s_j}\left( \left| \frac{\partial^4 \phi(\boldsymbol{\xi}_{ij}^{(3)})}{\partial s_i^4} \right| + \left| \frac{\partial^4 \phi(\boldsymbol{\xi}_{ij}^{(4)})}{\partial s_i^4} \right| \right)$$
$$+ \frac{\Delta s_i \Delta s_j}{8}\left( \left| \frac{\partial^4 \phi(\boldsymbol{\xi}_{ij}^{(5)})}{\partial s_i^2 \partial s_j^2} \right| + \left| \frac{\partial^4 \phi(\boldsymbol{\xi}_{ij}^{(6)})}{\partial s_i^2 \partial s_j^2} \right| \right) + \frac{\Delta s_j^3}{48\Delta s_i}\left( \left| \frac{\partial^4 \phi(\boldsymbol{\xi}_{ij}^{(7)})}{\partial s_j^4} \right| + \left| \frac{\partial^4 \phi(\boldsymbol{\xi}_{ij}^{(8)})}{\partial s_j^4} \right| \right).$$

Without loss of generality, we assume that $\Delta s = \Delta s_i = \Delta s_j$. Then, we have

$$\inf_{\Delta s} \frac{\sqrt{\varepsilon_0}}{\sqrt{D}\Delta s^2} + \frac{\Delta s^2}{6}\left| \frac{\partial^4 \phi(\boldsymbol{\xi}_{ij}^{(1)})}{\partial s_i^3 \partial s_j} \right| + \frac{\Delta s^2}{6}\left| \frac{\partial^4 \phi(\boldsymbol{\xi}_{ij}^{(2)})}{\partial s_i \partial s_j^3} \right| + \frac{\Delta s^2}{48}\left( \left| \frac{\partial^4 \phi(\boldsymbol{\xi}_{ij}^{(3)})}{\partial s_i^4} \right| + \left| \frac{\partial^4 \phi(\boldsymbol{\xi}_{ij}^{(4)})}{\partial s_i^4} \right| \right)$$
$$\tag{26}$$
$$+ \frac{\Delta s^2}{8}\left( \left| \frac{\partial^4 \phi(\boldsymbol{\xi}_{ij}^{(5)})}{\partial s_i^2 \partial s_j^2} \right| + \left| \frac{\partial^4 \phi(\boldsymbol{\xi}_{ij}^{(6)})}{\partial s_i^2 \partial s_j^2} \right| \right) + \frac{\Delta s^2}{48}\left( \left| \frac{\partial^4 \phi(\boldsymbol{\xi}_{ij}^{(7)})}{\partial s_j^4} \right| + \left| \frac{\partial^4 \phi(\boldsymbol{\xi}_{ij}^{(8)})}{\partial s_j^4} \right| \right). \tag{27}$$

By defining

$$\tau := 8\left( \left| \frac{\partial^4 \phi(\boldsymbol{\xi}_{ij}^{(1)})}{\partial s_i^3 \partial s_j} \right| + \left| \frac{\partial^4 \phi(\boldsymbol{\xi}_{ij}^{(2)})}{\partial s_i \partial s_j^3} \right| \right) + \left( \left| \frac{\partial^4 \phi(\boldsymbol{\xi}_{ij}^{(3)})}{\partial s_i^4} \right| + \left| \frac{\partial^4 \phi(\boldsymbol{\xi}_{ij}^{(4)})}{\partial s_i^4} \right| \right)$$
$$+ 6\left( \left| \frac{\partial^4 \phi(\boldsymbol{\xi}_{ij}^{(5)})}{\partial s_i^2 \partial s_j^2} \right| + \left| \frac{\partial^4 \phi(\boldsymbol{\xi}_{ij}^{(6)})}{\partial s_i^2 \partial s_j^2} \right| \right) + \left( \left| \frac{\partial^4 \phi(\boldsymbol{\xi}_{ij}^{(7)})}{\partial s_j^4} \right| + \left| \frac{\partial^4 \phi(\boldsymbol{\xi}_{ij}^{(8)})}{\partial s_j^4} \right| \right),$$

we have the following form of (26)

$$\inf_{\Delta s} \frac{\sqrt{\varepsilon_0}}{\sqrt{D}\Delta s^2} + \frac{\tau \Delta s^2}{48}. \tag{28}$$

By taking derivative w.r.t. $\Delta s$, we have the minimizer

$$\Delta s = \left( \frac{48\sqrt{\varepsilon}}{\sqrt{D}\tau} \right)^{1/4},$$

which gives the optimal

$$\frac{2\sqrt{3\tau}\varepsilon^{1/4}}{3D^{1/4}}.$$

Therefore, we have

$$\left| \mathbb{E}\left[ \sum_{m=1}^{M} q_{m,k} a_{m,i} a_{m,j} \widehat{h}_m''(\boldsymbol{A}\boldsymbol{s}_\ell) \right] \right| \leq \frac{2\sqrt{3\tau}\varepsilon^{1/4}}{3D^{1/4}} \leq \frac{2\sqrt{96 C_d}\varepsilon^{1/4}}{3D^{1/4}}, \tag{29}$$

since $\tau \leq 32 C_d$.

By setting $\varepsilon = DM\nu^2 + 4DMB^4 C_x^2 \sqrt{\frac{R}{N}} + 4DMB^4 C_x^2 \sqrt{\frac{2\log(4/\delta)}{N}}$, we have

$$\left| \mathbb{E}\left[ \sum_{m=1}^{M} q_{m,k} a_{m,i} a_{m,j} \widehat{h}_m''(\boldsymbol{A}\boldsymbol{s}_\ell) \right] \right| \leq \frac{8\sqrt{6 C_d}}{3}\left( M\nu^2 + 4MB^4 C_x^2 \sqrt{\frac{R}{N}} + 4MB^4 C_x^2 \sqrt{\frac{2\log(4/\delta)}{N}} \right)^{1/4}.$$
$$\tag{30}$$

## D.3 Combining the Results

Now we put together the estimation in (24) and (30). Since the $\ell_2$ norm is upper bounded by $\ell_1$ norm, we have the following

$$\mathbb{E}\left[\left\|\boldsymbol{Q}^\top \odot \boldsymbol{B}\widehat{\boldsymbol{h}}''(\boldsymbol{As})\right\|_2^2\right] = O\left(C_\phi\sqrt{M}\left(\nu^2 + B^4C_x^2\sqrt{\frac{R}{N}} + B^4C_x^2\sqrt{\frac{2\log(4/\delta)}{N}}\right)^{1/2}\right),$$

$$\leq O\left(C_\phi\sqrt{M}\nu + C_\phi B^2 C_x\left(\sqrt{\frac{R}{N}} + \sqrt{\frac{2\log(4/\delta)}{N}}\right)^{1/2}\right),$$

$$\leq O\left(C_\phi\sqrt{M}\nu + \frac{C_\phi B^2 C_x\left(\sqrt{R} + \sqrt{2\log(4/\delta)}\right)^{1/2}}{N^{1/4}}\right), \qquad (31)$$

due to the fact that $\sqrt{a+b} \leq \sqrt{a} + \sqrt{b}$ for $a, b \geq 0$. The above can be further written as

$$\mathbb{E}\left[\left\|\widehat{\boldsymbol{h}}''(\boldsymbol{As})\right\|_2^2\right] = O\left(\frac{C_\phi\sqrt{M}\nu}{\sigma_{\min}^2(\boldsymbol{Q}^\top \odot \boldsymbol{B})} + \frac{C_\phi B^2 C_x\left(\sqrt{R} + \sqrt{2\log(4/\delta)}\right)^{1/2}}{\sigma_{\min}^2(\boldsymbol{Q}^\top \odot \boldsymbol{B})N^{1/4}}\right), \qquad (32)$$

which completes the proof.