# OpenReview forum: "Provable Subspace Identification Under Post-Nonlinear Mixtures"
_NeurIPS.cc/2022/Conference — NeurIPS 2022 Accept_

### Official Review · Reviewer_TZbZ · 2022-07-03

**Rating:** 6
**Confidence:** 4
**Soundness:** 3 good
**Presentation:** 3 good
**Contribution:** 3 good

**Summary:**

This paper tackles a very challenging problem: blind source separation under nonlinear mixing. The authors made a reasonable simplification that the nonlinearity is element-wise, to make the problem more tractable. The authors proposed a null-space identification algorithm to tackle the nonlinearity and the mixing matrix. Theoretical analysis on the identifiability properties were also provided to justify the proposed algorithm, together with convincing simulation results. The paper is very interesting and sheds lights on related independent component analysis problems.

**Questions:**

1) In Section 6, the reconstruction network r is required to have the same structure of the transformation network f. How important is this requirement? What if a different architecture is used for r?

2) How is the regularization parameter \lambda determined? Is the result sensitive to the setting of \lambda?



**Limitations:**

The element-wise nonlinearity function is assumed throughout the paper. Can the proposed null-space algorithm be extended to arbitrary nonlinear mixing in equation (2)?

**Strengths And Weaknesses:**

Strengths:
1) The authors proposed a novel null-space identification approach to simultaneously tackle the nonlinearity and the mixing matrix.
2) An autoencoder reconstruction loss is included to force the transformation invertible. Moreover, end-to-end optimization can be carried out in the block coordinate descent sense.
3) The theory is elegant and the experimental result is promising.

Weakness:
1) The algorithm could identify the null space of the linear mixing matrix and a nonlinear transform. It is still unclear to me how to guarantee  the recovery of the original signals from the null space, as there is still linear transform ambiguity there. As a comparison, the classic infomax algorithm (Bell and Sejnowski 1995) directly estimates a linear unmixing matrix based on a mutual information maximization criterion.

2) The requirement for a dense null-space matrix Q seems unnecessary for me. Take an extremely simple case, where the mixing matrix A is the first K columns of a M*M identity matrix, and the nonlinearity function g is an identity mapping. In this case, Q should be in the null space of A, which means that the first K rows of Q must be zero, which violates the dense requirement.

3) It would be nice if the authors can compare this proposed method with other nonlinear ICA / BSS methods, for example, classical methods reviewed in the following paper:
Deville Y, Duarte LT. An overview of blind source separation methods for linear-quadratic and post-nonlinear mixtures. In International Conference on Latent Variable Analysis and Signal Separation 2015 Aug 25 (pp. 155-167). Springer.

---

> ### Author Response · Authors · 2022-08-01
> **Response to Reviewer TZbZ**
>
> We thank the reviewer for his/her appreciation of our work.
>
> **[Weakness]**
> 1. Note that if g is “removed”, the remaining problem for identifying s becomes a well-studied linear mixture model-based BSS problem. Then, by exploiting properties of s and/or $A$, e.g., quasi-stationarity, nonnegativity, and boundedness, s can be identified by existing tools. We mentioned this in line 26-29 of the paper. We will make sure that this point is more articulated in the paper.
> The work in [Bell and Sejnowski 1995] is somewhat not directly comparable with our method, as [Bell and Sejnowski 1995] assumed that the nonlinear function’s function classes (e.g., $g(\cdot)$ is a logistic function, sigmoid function or tanh function) are known. In our case, such knowledge is not available. In addition, [Bell and Sejnowski 1995] assumed that the sources are statistically independent, but our method for removing $g(\cdot)$ does not need to use this assumption.
>
> 2. We should clarify that the dense constraint on $Q$ is not for constructing a basis of $\text{null}(A^\top)$, but an additional constraint to serve for the purpose of deriving the model identifiability. Note in the proof of Thm. 1 that we constrain $Q$ to be dense such that its Kruskal rank is at least 1. Hence, we are not only asking for a basis of $\text{null}(A^\top)$ but also a dense basis—which always exists.
>
> 3. Thanks for the suggestion. We have added [Ziehe et al. 2003] (PNL-ICA) and [Khemakhem et al. 2020] (nICA). The results are as follows and will be updated in the final version.
> Method | Raw | PCA | AE | Proposed | nICA | PNL-ICA
> -------|--------|-------|--------|-------|--------|-------
> SVM   | 0.46±0.05 (0.43)  | 0.43±0.02 (0.41) | 0.43±0.03 (0.40) | **0.40±0.03 (0.35)** | 0.44±0.02 (0.43) | 0.46±0.05 (0.38)
> LR   | 0.47±0.04 (0.42) | 0.45±0.04 (0.41) | 0.46±0.03 (0.41) | **0.39±0.03 (0.35)** | 0.43±0.03 (0.39) | 0.44±0.03 (0.39)
>
> [Ziehe et al. 2003] (PNL-ICA) Ziehe, Andreas, et al. "Blind separation of post-nonlinear mixtures using linearizing transformations and temporal decorrelation." The Journal of Machine Learning Research 4 (2003): 1319-1338.
>
> [Khemakhem et al. 2020] (nICA) Khemakhem, Ilyes, et al. "Variational autoencoders and nonlinear ICA: A unifying framework." International Conference on Artificial Intelligence and Statistics. PMLR, 2020.
>
> **[Questions]**
> 1. In theory, it does not require $f(\cdot)$ and $r(\cdot)$ to have the same architecture, as long as $f(\cdot)$ and $r(\cdot)$ are expressive enough to represent $g^{-1}(\cdot)$ and $g(\cdot)$, respectively. In practice, it seems natural to use the same architecture as $g^{-1}(\cdot)$ and $g(\cdot)$ should be similarly complex.
>
> 2. The parameter was not fine-tuned and the method seems to be reasonably robust to this parameter. For the simulations, there is a relatively wide range of lambdas (e.g., lambda in $[10^{-5}, 10^{-3}]$) that give similar results. Our principle of choosing lambda was to set a lambda that is not too large to “overwhelm” the first term in the objective function, as the first term is more important for enforcing the functional equations. We will add a comment in the revised version to discuss the choice of lambda. This works well for the simulations in Figs. 1-2.
> In real data experiments, an additional way to tune the parameter is to make use of the downstream tasks. For example, in our case, the downstream task is a classification task. Hence, we used the validation set to tune the parameter.
>
> **[Limitations]**
>
> Our work focuses on the PNL model, which has this nice element-wise nonlinearity to exploit. Since in Eq (2), there is no linear mixing part, i.e., $As$, then the method of looking for $\text{null}(A^\top)$ may not be applicable.

---

> > ### Comment · Reviewer_TZbZ · 2022-08-07
> > **Final comments**
> >
> > Thanks authors for responding to my questions.
> > My rating of this work remains unchanged.

---

### Official Review · Reviewer_Gx16 · 2022-07-11

**Rating:** 5
**Confidence:** 3
**Soundness:** 3 good
**Presentation:** 2 fair
**Contribution:** 2 fair

**Summary:**

The authors prove that the existence of a nontrivial  null space  associated with the underlying mixing system suffices to guarantee identification/removal of the unknown nonlinearity. Consequently, a simple learning criterion is proposed. A sample complexity analysis is offered to characterize the performance of the proposed approach under realistic settings. An optimization algorithm using BCD is proposed and evaluated through numerical experiments.

**Questions:**


Question 1:  In Eq. (5c), to avoid degenerated solutions $f_m(\cdot)=c$, the functions $f_m(\cdot)$ is assumed to be invertible. Is this constraint too stringent?

Question 2:  In Eq. (5d), is the non-sparsity constraint $\|Q\|_0=MD$ too restricted?

Question 3: In Section 5 "Sample Complexity Analysis", Theorem 2 starts with "Sample Complexity" but does not give an explicit bound on the sample complexity. It is suggested to give an explicit expression of the sample complexity.

**Limitations:**

Yes, the limitations of the proposed approach (i.e., the efficiency, the non-existence of orthogonal component, and the non-convexity of the proposed criterion) have been adequately discussed in the last paragraph.

**Strengths And Weaknesses:**

[Strengths]

Strength 1: It is proved that the existence of a nontrivial null space associated with the underlying mixing system suffices to guarantee identification/removal of the unknown nonlinearity.

Strength 2: A sample complexity analysis is provided to characterize the performance of the proposed approach under realistic settings.


[Weaknesses]

Weakness 1: As each dimension is modeled with an individual neural network, the PNL model would incur high computational cost when the data feature dimension M is large.

Weakness 2: There are examples where the A matrix may not have a nontrivial orthogonal complement such that the proposed approach fails.

---

> ### Author Response · Authors · 2022-08-01
> **Response to Reviewer Gx16**
>
> We would like to thank the reviewer for the positive evaluation.
>
> **[Weakness]**
> 1. Computational Cost: Indeed, the nonlinear mixture models may have an interesting tradeoff between computational cost and the easiness of identifiability. The PNL is in general easier to identify than unstructured nonlinear mixtures, but the individual nonlinear distortion on each dimension may cause higher computational complexity. Our conjecture is that using multiple computing agents and parallel computing could reduce the computational time, which would be an interesting topic to study in the future.
> 2. Cases where $A$ does not have an orthogonal complement: Indeed, as we pointed out in our conclusion, this is a limitation of the approach. Nonetheless, in many applications a tall $A$ is commonly seen, e.g., speech separation, hyperspectral unmixing, and EEG processing. For cases where $A$ is fat or square, one indeed would need to resort to other properties for nonlinearity removal and model identification.
>
> **[Questions]**
> 1. Note that under the PNL models, the generative $g(\cdot)$ is invertible; see [Taleb and Jutten 1999, Ziehe et al. 2003, Deville and Duarte 2015.]. Otherwise, it is not possible to “cancel” $g(\cdot)$. Hence, asking $f(\cdot)$ to be invertible is natural, as $f(\cdot)$ acts as the inverse of $g(\cdot)$. Our experience is that, in practice, promoting $f(\cdot)$ to be invertible is not very hard. One can use invertible neural networks such as the normalizing flow. One can also use our autoencoder based implementation.
> 2. Our experience is that promoting a dense $Q$ is not hard. As we discussed in the paper, if $Q$ is randomly initialized using continuous distributions, then after a continuous transformation, the transformed $Q$ is still following a certain joint continuous distribution, which means $Q$ is dense with probability one. In practice, the proposed iterative algorithm for updating $Q$ can be considered as a continuous transformation (in particular, the SVD-based projection uses orthogonal iterations which are just linear transformations). Hence, in practice, when we randomly initialize $Q$, we never encounter a sparse $Q$ in our algorithm output.
> 3. Theorem 2 characterizes how the nonlinearity removal performance scales with the number of samples N. The performance is inversely proportional to the sample size $N^{1/4}$ (see the second term of (14)). In addition, the performance is also determined by the modeling error denoted by $\nu$ (see the first term of (14) and the definition of $\nu$ in Assumption 1), which is not affected by the sample size but solely related to how expressive the learning function class is. We were referring this characterization as "sample complexity". In the revised version, we will change the term to "finite-sample identifiability analysis".
>
> [Taleb and Jutten 1999] Taleb, Anisse, and Christian Jutten. "Source separation in post-nonlinear mixtures." IEEE Transactions on signal Processing 47.10 (1999): 2807-2820.
>
> [Ziehe et al. 2003] (PNL-ICA) Ziehe, Andreas, et al. "Blind separation of post-nonlinear mixtures using linearizing transformations and temporal decorrelation." The Journal of Machine Learning Research 4 (2003): 1319-1338.
>
> [Deville and Duarte 2015] Deville, Yannick, and Leonardo Tomazeli Duarte. "An overview of blind source separation methods for linear-quadratic and post-nonlinear mixtures." International Conference on Latent Variable Analysis and Signal Separation. Springer, Cham, 2015.

---

> > ### Comment · Reviewer_Gx16 · 2022-08-08
> > **Final comment**
> >
> > Many thanks to the authors for their response.
> > In the response, Weaknesses 2 and 3, and Questions 1, 2, and 3 have been clearly explained, whereas Weakness 1 seems not well addressed. Based on the feedback, I would like to keep my rating of this work unchanged.

---

### Official Review · Reviewer_zio7 · 2022-07-12

**Rating:** 7
**Confidence:** 4
**Soundness:** 4 excellent
**Presentation:** 4 excellent
**Contribution:** 3 good

**Summary:**

The authors propose an apporach for provably identifying subspaces under post-nonlinear mixtures. The main idea is the formulation of subspace identification problem as nonconvex optimization problem where one learns both a nullspace matrix and an invertible map that removes non-linearities. Theoretical results show that subspace identification is possible under certain assumptions that involve degrees of freedom of latent variables and the  of the nullspace of the linear mixing matrix A.

**Questions:**

1) It is not clear why $\mathbf{Q}$ should be dense. Why the orthogonality constraint is not sufficient to avoid trivial solutions?
2) It seems that the non-linear map should be defined as : $g: \mathbb{R}^M \rightarrow \mathbb{R}^M $. Can the results be generalized to mappings $g: \mathbb{R}^M \rightarrow \mathbb{R}^N$ for any N?

**Limitations:**

Yes

**Strengths And Weaknesses:**

Strengths:
1) The paper is easy to follow and well-written.
2)  The approach is very interesting and the reported results are novel and quite insightful. It is quite surprising that properties of non-linear function g do not play any role in the conditions for subspace identification and removal of non-linear mapping.
3) The authors provide simulated and a real EEG experiment showing promising results for their method.

Weakness:
1) The optimization problem is non-convex and might be hard to guarantee convergence to optimal f and Q.
The authors do not provide any insight as to  possible directions regarding the analysis of the optimization problem.
2) The proposed method compares with very standard classifiers e.g. SVM and logistic regression. It would be interesting to compare the performance  more state-of-the-art approaches e.g.  a DNN classifier.

---

> ### Author Response · Authors · 2022-08-01
> **Response to Reviewer zio7**
>
> We would like to thank the reviewer for the positive comments on our work.
>
> **[Weakness]**
>
> 1. **[Optimization]** Indeed, as we agreed in the conclusion section, optimization of the formulation is hard. The formulated problem has a nonconvex objective function. There is also a manifold constraint that makes the constraints nonconvex. Under reasonable conditions, the algorithm converges to a stationary point of the formulated problem, under standard block coordinate descent (BCD) analytical tools. Nonetheless, converging to a stationary point is not really satisfactory as the identifiability is attained at the global optima. Although empirically we observe that our optimization strategy often successfully removes the nonlinear functions $g(\cdot)$,  the theoretical underpinning  of the global optimality-attaining properties of the algorithm may not be a trivial task and perhaps deserves a standalone study. Nonetheless, we will comment on the optimization procedure and the convergence properties as well as the challenges in analysis in the revised version.
>
> 2. **[Evaluation with SVM and Logistic Regression]** The rationale behind the evaluation follows those in [Wang et al. 2015, Hyvarinen et al. 2016, Chen et al. 2020]. To be specific, if one believes that the nonlinear unmixing stage learns informative latent representations, then the representations should be “friendly” for simple classifiers to work.
> Following the comment of the reviewer, we have used a more complex classifier (i.e., fully connected DNN). The DNN has 2 layers and each has 256 neurons. The neurons are ReLU activation functions. The architecture is tuned over a validation set. One can see that our method still outperforms the DNN classifier.
> The result is as follows:
> Method | Proposed-SVM | Proposed-LR | DNN
> -------|--------|-------|-------
> ERR   | 0.40±0.03 (0.35) | 0.39±0.03 (0.35) | 0.43±0.02 (0.41)
>
> [Wang et al. 2015] Wang, Weiran, et al. "On deep multi-view representation learning." International conference on machine learning. PMLR, 2015.
>
> [Hyvarinen et al. 2016] Hyvarinen, Aapo, and Hiroshi Morioka. "Unsupervised feature extraction by time-contrastive learning and nonlinear ICA." Advances in neural information processing systems 29 (2016).
>
> [Chen et al. 2020] Chen, Ting, et al. "A simple framework for contrastive learning of visual representations." International conference on machine learning. PMLR, 2020.
>
>
> **[Questions]**
> 1. The reviewer is right that the orthogonality constraint is enough to guarantee that the $Q$ matrix is not degenerated. However, a dense $Q$ is required to derive the model identifiability. In particular, we need $Q$ to be a dense matrix such that its Kruskal rank is at least 1 (see line 189-195 of proof of Thm. 1).  We will add a comment to point it out in the revised version.
> 2. By the nature of post-nonlinear mixture, $g(\cdot)$ is always an $M$ to $M$ function. Changing $g(\cdot)$ to an $M$ to $N$ function may break the post-nonlinear mixture structure, and may need some nontrivial redesign of the entire approach.

---

> > ### Comment · Reviewer_zio7 · 2022-08-08
> > **Final Comment**
> >
> > I want to thank the authors for their response to my review and the clarifications they provided. I think that the requirement of Q to be dense should be emphasized and better explained in the manuscript and I encourage the authors to better explain that part in the revised version of the paper. All in all, I believe that this is a good paper and I will keep my score unchanged.

---

### Official Review · Reviewer_3LT3 · 2022-07-14

**Rating:** 8
**Confidence:** 3
**Soundness:** 4 excellent
**Presentation:** 4 excellent
**Contribution:** 4 excellent

**Summary:**

The paper introduces a novel method to identify post-nonlinear mixture models under mild assumptions by removing the nonlinear transformations, after which standard linear mixture model identification methods can be employed to identify the latent components. The problem is formulated as a constraint optimisation problem that can be uniquely solved given infinite data. Additionally a finite sample analysis with probability bounds is derived. An algorithmic implementation of the method is provided and evaluated on synthetic examples for both independent and dependent components to support the theoretical claims.


**Questions:**

- eq(2) vs. eq(3): do I understand correctly that the property M > K (‘A is a tall matrix’, l.125) is what makes the system in (3) identifiable whereas (2) is not? Then why is the relation in eq(9) of the form K(K+1) > 2M? Or should both hold?


**Limitations:**

No concerns on this aspect.

**Strengths And Weaknesses:**

strengths:
+ very interesting contribution to a well established and frequently encountered problem
+ high quality paper, well written with good explanations, despite the fairly complex/abstract techniques employed
+ significant and surprising identifiability result
+ potential to lead to relevant novel applications in various fields

weakness:
- experimental evaluation could/should have been stronger (perhaps move more proof details to the supplement)

As far as I can tell results are sound and novel, although I have to admit the subject is somewhat outside of my area of expertise, hence reduced confidence. Nevertheless: clear accept.

minor remarks:
- abstract reads a bit like an introduction, but ok
- 28: typo ’non(n)egativty’
- 60: missing word ‘.. those (that) are ..’
- 149: missing word ‘.. (are) all locally …’
- 287: Fig.7 => Fig.1
- 301: Fig.7 => Fig.2

---

> ### Author Response · Authors · 2022-08-01
> **Response to Reviewer 3LT3**
>
> We thank the reviewer for his/her appreciation of our work.
>
> **[Weakness]**
>
> We will add two nonlinear ICA baselines to enrich the real-experiment experiment results [Ziehe et al. 2003, Khemakhem et al. 2020]. ICA based methods require that the latent components are statistically independent, which is often a stringent condition. As a consequence, one can see that the results of using ICA-based methods are less promising for the EEG experiment.
>
> Method | Raw | PCA | AE | Proposed | nICA | PNL-ICA
> -------|--------|-------|--------|-------|--------|-------
> SVM   | 0.46±0.05 (0.43)  | 0.43±0.02 (0.41) | 0.43±0.03 (0.40) | **0.40±0.03 (0.35)** | 0.44±0.02 (0.43) | 0.46±0.05 (0.38)
> LR   | 0.47±0.04 (0.42) | 0.45±0.04 (0.41) | 0.46±0.03 (0.41) | **0.39±0.03 (0.35)** | 0.43±0.03 (0.39) | 0.44±0.03 (0.39)
>
> In the revision, we will add these results in the supplementary material.
>
> [Ziehe et al. 2003] (PNL-ICA) Ziehe, Andreas, et al. "Blind separation of post-nonlinear mixtures using linearizing transformations and temporal decorrelation." The Journal of Machine Learning Research 4 (2003): 1319-1338.
>
> [Khemakhem et al. 2020] (nICA) Khemakhem, Ilyes, et al. "Variational autoencoders and nonlinear ica: A unifying framework." International Conference on Artificial Intelligence and Statistics. PMLR, 2020.
>
> **[Questions]**
>
> Indeed, **both should hold** in order to identify the nonlinear generative functions. First, $M>K$ is required for the existence of a null space such that there is a solution to (5). But the existence of a solution is not sufficient to identify the latent model. Moreover, Eq. (9) is needed to make sure that matrix $\widehat{A}$ in (10) is full Kruskal rank. With such a condition, we are able to pin down the nonlinear generative functions. The insight here is that $M$ should be larger than $K$, but not overly large. The reason is that there are $M$ unknown $g_i(\cdot)$ to be identified, and thus a very large $M$ would make the problem harder to deal with as more unknowns are present. We will add a comment to reflect the above discussion.

---

### Meta-Review · Area_Chair_rzmY · 2022-08-24

**Recommendation:** Accept
**Confidence:** Certain

**Metareview:**

The post-nonlinear (PNL) mixture model has been studied for a while in BSS community.  This paper presents a method to identity nonlinearity under mild conditions so that after removing the nonlinear transformations, a method for source separation  from a linear mixture is employed. All of reviewers agree that the paper is well written and has a solid contribution in BSS. Two important contributions are: (1) it is proved that the existence of a non-trivial null space associated with the underlying mixing systems suffices to guarantee identification/removal of the unknown nonlinearity; (2) a sample complexity analysis is provided to characterize the performance of the proposed approach. A downside is in weak experiments, which has been improved during the author rebuttal period. Therefore, I am pleased to suggest the paper to be accepted.


**Award:**

No

---

### Decision · Program_Chairs · 2022-09-14

Accept